# Instruction-Following LLMs for Time Series Prediction: A Two-Stage Multimodal Approach

## Abstract

We introduce **Text-Informed Time Series Prediction (TITSP)**, an innovative multimodal framework that integrates textual knowledge with temporal dynamics using Large Language Models (LLMs). TITSP employs a two-stage process that bridges numerical data with rich contextual information for enhanced forecasting accuracy and interpretability. In the first stage, we present **AutoPrompter**, which captures temporal dependencies from time series data and aligns them with semantically meaningful text embeddings. In the second stage, these aligned embeddings are refined by incorporating task-specific textual instructions through LLM. We evaluate TITSP on several multimodal time series prediction tasks, demonstrating substantial improvements over state-of-the-art baselines. Quantitative results reveal significant gains in predictive performance, while qualitative analyses show that textual context enhances interpretability and actionable insights. Our findings indicate that integrating multimodal inputs not only improves prediction accuracy but also fosters more intuitive, user-centered forecasting.

## 1 Introduction

Time series prediction is critical in fields such as finance, healthcare, and climate science, where timely and accurate forecasts drive informed decision-making. Traditional time series forecasting pipelines typically involve three key stages: data preprocessing, model selection, and performance evaluation.

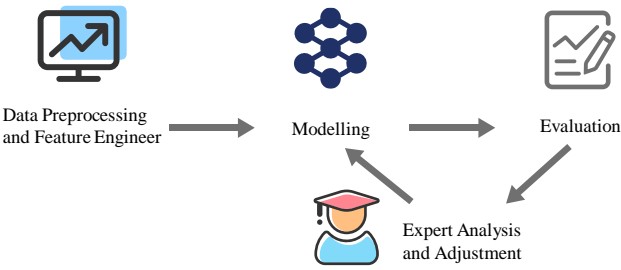

Figure 1: The ideal complete process of Time-series Prediction

However, methods like ARIMA (Shumway et al., 2017), while foundational, often struggle to capture the complex non-linear patterns and long-range dependencies present in real-world datasets. With the rise of deep learning, models such as Long Short-Term Memory (LSTM) networks (Siami-Namini et al., 2019) and Convolutional Neural Networks (CNNs) (Wang et al., 2019) have shown significant improvements in modeling these complexities. However, they remain constrained by their reliance on numerical data alone, which limits their ability to integrate external contextual information—such as expert insights or macroeconomic events—that could enhance forecast accuracy and interpretability. This shortcoming becomes especially problematic in chaotic or highly volatile systems, such as financial markets or patient health monitoring.

Recent advancements in Transformer-based architectures Ilbert et al.; Zhou et al. (2021); Wen et al. (2022) and Large Language Models (LLMs) (Gruver et al., 2024; Jin et al., 2023) have enhanced the ability to capture long-range dependencies in time series data. Yet, these models also face limitations when applied in isolation, often missing out on crucial domain-specific insights provided through other modalities, such as text. In practical scenarios, domain experts may wish to provide instructions or insights that can guide the forecasting process, but existing models do not easily accommodate such interactions.

To address these challenges, we propose **Text-Informed Time Series Prediction** (TITSP), a novel two-stage framework that combines the strengths of deep learning models for time series prediction with the contextual richness of domain-specific textual inputs. TITSP first captures temporal dependencies using numerical time series data and then refines these predictions by incorporating task-specific textual instructions through a Large Language Model. This integration allows the model to generate more accurate, context-aware, and interpretable predictions. Our extensive experiments demonstrate that TITSP significantly outperforms state-of-the-art models, particularly in scenarios where expert input is essential.

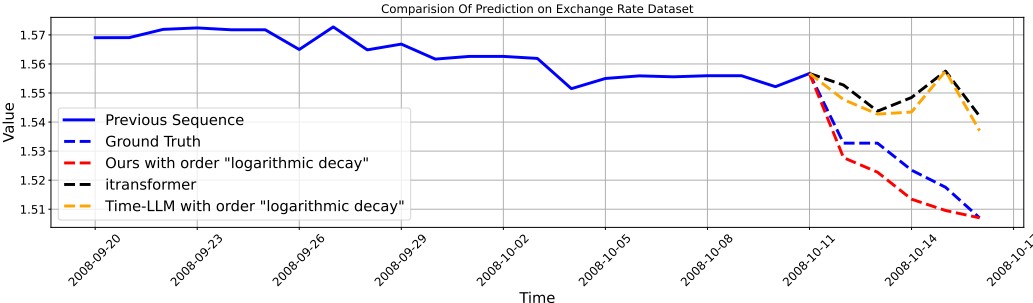

Figure 2: Interactive Prediction of GBP/USD Exchange Rate during the 2008 Financial Crisis compared with other methods.

Figure 2 demonstrates the effectiveness of TITSP in predicting the GBP/USD exchange rate during the 2008 financial crisis. It highlights how expert textual inputs can enhance predictions in volatile environments. Additionally, Appendix J provides details on why a finance expert might offer insights into the logarithmic decay observed on October 11, 2008.

The remainder of the paper is structured as follows. In Section 2 , we discuss related work, covering existing approaches in time series prediction and multimodal learning. Section 3 formalizes the problem statement and the key challenges we aim to address. In Section 4, we describe our proposed methodology, providing a detailed explanation of the architecture and the rationale behind its design. Section 5 presents the experimental results, showcasing the effectiveness of our approach through evaluations on multiple benchmark datasets.

The code to reproduce the results of this paper is included in the supplementary material and will be made publicly available upon acceptance.

## 2 RELATED WORKS

### 2.1 TIME SERIES FORECASTING APPROACHES

Time series forecasting began with classical models like ARIMA (Box et al., 2015), which are effective for linear patterns but struggle with non-linearities common in real-world data (Chatfield, 2000). Feature extraction methods like *tsfresh* (Christ et al., 2018) and machine learning algorithms such as Random Forests (Breiman, 2001) and Support Vector Machines (Vapnik, 1998) improved predictive accuracy by capturing non-linear dependencies.

The adoption of deep learning models, particularly RNNs (Rumelhart et al., 1986), LSTMs (Hochreiter & Schmidhuber, 1997), and Transformers (Vaswani et al., 2017), has further advanced the field by automatically learning complex temporal patterns (Wen et al., 2017). Following the sucess of

transformer, works including TimeXer (Wang et al., 2024), itransformer (Liu et al., 2023), PatchTST (Nie et al., 2022) and Informer (Zhou et al., 2021) are designed to address time-series prediction problems. Despite these advancements, most approaches rely solely on time series data. Our work **diverges by integrating time series with textual context via Large Language Models (LLMs)**, offering a richer, more contextual understanding of the data.

## 2.2 MULTIMODAL LEARNING IN TIME SERIES PREDICTION

Integrating time series data with unstructured text has become a key strategy for enhancing forecasting accuracy through multimodal approaches that leverage diverse data types for richer context.

**Advances in Model Architectures and Joint Prediction** Zhang et al. (2024) introduced the Dual-Adapter model to optimize time series representation by balancing textual and temporal information. Building on this, Liu et al. (2024b) proposed UniTime, a unified model leveraging NLP for enhanced time series forecasting across domains, while Liu et al. (2024a) explored LLMs for dynamical systems, showcasing their adaptability. Surveys such as (Liang et al., 2024; Deldari et al., 2022) offer taxonomies of multimodal learning, identifying gaps in self-supervised methods for multimodal and temporal data. The Multi-Modal Forecaster by Kim et al. (2024) jointly predicts time series and text, highlighting the potential of cohesive multimodal frameworks. Similarly, Cheng et al. (2024) introduced a prompt-based multimodal framework focused on classification. Recent advancements, such as (Jia et al., 2024) and (Jin et al., 2023), further integrate time series and text using LLMs, enhancing predictions through textual cues.

**Existing Gaps and Novel Contributions.** While recent works have advanced the integration of time series and text, a key gap remains in **instruction-based time series forecasting**. Our method addresses this by introducing a novel framework for text instruction-based forecasting, leveraging the interplay between textual instructions and time series data for richer, interactive predictions. Additionally, we present a tailored methodology to benchmark such tasks, enabling applications in "what-if" scenarios and dynamic, user-interactive forecasting.

## 2.3 CONTRIBUTIONS OF OUR WORK.

Building on recent advancements in multimodal time series forecasting, our work makes the following key contributions:

- **Data Generation Workflow:** We propose a novel data generation workflow that enables the evaluation of instruction-based time series forecasting. This workflow facilitates the testing of model applicability in scenarios that require adherence to specific instructions, a previously underexplored area.
- **Innovative Two-Stage Approach:** We introduce a two-stage model that effectively integrates both temporal data and textual instructions. This design captures contextual information while adhering to instruction-based predictions, leading to enhanced performance.
- **Comprehensive Ablation Studies:** We conduct ablation studies to validate each step of the proposed approach, demonstrating the importance and rationale behind each component.
- **Comparative Evaluation:** We benchmark our method against state-of-the-art models for multimodal time series and text forecasting, showing competitive or superior performance across key metrics (both in terms of a new metrics called compliance rate and in terms of MSE).

These contributions address a significant gap in instruction-based forecasting, offering a structured and interpretable solution.

## 3 PROBLEM DEFINITION

Let $\mathbf{X} = \{\boldsymbol{x}_1, \boldsymbol{x}_2, \ldots, \boldsymbol{x}_T\}$ denote a multivariate time series, where $\boldsymbol{x}_t \in \mathbb{R}^d$ represents the $d$-dimensional vector at time step $t \in \{1, 2, \ldots, T\}$. Traditional time series forecasting aims to predict future values $\hat{\mathbf{X}} = \{\hat{\boldsymbol{x}}_{T+1}, \hat{\boldsymbol{x}}_{T+2}, \ldots, \hat{\boldsymbol{x}}_{T+H}\}$, where $H$ is the forecasting horizon, using only the historical observations $\mathbf{X}$.

In real-world applications, textual information often provides critical context or specific instructions that can influence the expected trajectory of the time series. Let $\mathbf{S} = s_1, s_2, \ldots, s_n$ be a sequence of $n$ tokens representing a textual description or instruction, where $s_i \in \mathcal{V}$ and $\mathcal{V}$ is a fixed vocabulary. This text can convey information about **future events**, **conditions**, or **hypothetical scenarios** that could affect the future behavior of the time series.

Our objective is to design a multimodal framework that integrates both the time series $\mathbf{X}$ and the text $\mathbf{S}$ to produce a more **informed** forecast $\hat{\mathbf{X}}^{(\mathbf{S})} = \{\hat{\boldsymbol{x}}_{T+1}^{(\mathbf{S})}, \hat{\boldsymbol{x}}_{T+2}^{(\mathbf{S})}, \ldots, \hat{\boldsymbol{x}}_{T+H}^{(\mathbf{S})}\}$, conditioned on both the historical data and the information in $\mathbf{S}$. Formally, the forecasting function is defined as:

$$\hat{\mathbf{X}}^{(\mathbf{S})} = f(\mathbf{X}, \mathbf{S}; \theta) \tag{1}$$

where $f(\cdot, \cdot; \theta)$ is a function parameterized by $\theta$, capturing the relationships between the time series and the text. The goal is to learn the parameters $\theta$ to minimize the forecasting error:

$$\theta^* = \arg\min_\theta \mathcal{L}\left(\hat{\mathbf{X}}^{(\mathbf{S})}, \mathbf{X}^{\text{true}}\right) \tag{2}$$

where $\mathcal{L}$ is a loss function that measures the discrepancy between the forecasted values $\hat{\mathbf{X}}^{(\mathbf{S})}$ and the true future values $\mathbf{X}^{\text{true}} = \{\boldsymbol{x}_{T+1}^{\text{true}}, \boldsymbol{x}_{T+2}^{\text{true}}, \ldots, \boldsymbol{x}_{T+H}^{\text{true}}\}$.

## 4 PROPOSED METHODOLOGY

### 4.1 DATA GENERATION

To train and evaluate our multimodal forecasting model, we generate a synthetic dataset that integrates time series data with textual instructions. The goal is to modify forecasted time series values based on text descriptions, simulating real-world scenarios where instructions influence future predictions.

The data generation process consists of the following steps:

- **Base Time Series Creation:** We generate multivariate time series $\mathbf{X} = \{\boldsymbol{x}_1, \boldsymbol{x}_2, \ldots, \boldsymbol{x}_T\}$ using standard generators (e.g., sine waves, real-world data including financial, electricity data among others). The full description of all the base time series used for the generation is provided in Appendix C.
- **Textual Instruction Integration:** For each time series, we generate corresponding textual instructions $\mathbf{S}$, such as *"increase," "decrease," or "stabilize,"*, ..., which provide guidance on how to modify future values.
- **Forecast Modification Based on Text:** Forecasted values $\hat{\mathbf{X}}^{(\mathbf{S})} = \{\hat{\boldsymbol{x}}_{T+1}^{(\mathbf{S})}, \ldots, \hat{\boldsymbol{x}}_{T+H}^{(\mathbf{S})}\}$ are adjusted based on the instructions.

As example, for the instruction **"trend up"**, the forecast is modified by applying a linear increment to the last observed value $x_T$:

$$\hat{x}_{T+h}^{(\mathbf{S})} = x_T + A \times h, \quad \text{where} \quad h \in \{1, 2, \ldots, H\} \tag{3}$$

In this expression, $A$ is a constant controlling the upward trend. In the data generation, $A$ evolves with time while being positive to add complexity to the generative model, as seen in Figure 3, where some non-linear transformations may be observed for linear growth. Further details on the instructions and their corresponding mathematical modifications are provided in Appendix 6.

To demonstrate the realism of the synthetic dataset, we present several example scenarios where textual instructions are used to modify the base time series. Each scenario shows the base time series and its modified forecasted values, highlighting how the text influences the trajectory of future predictions. Figure 3 shows three different scenarios: *"Linear Growth"*, *"Exponential Downward"*, and *"Logarithmic Decay"*. These examples illustrate the diversity of the dataset and how textual instructions are effectively incorporated into the time series predictions. More figures are listed in Appendix C.

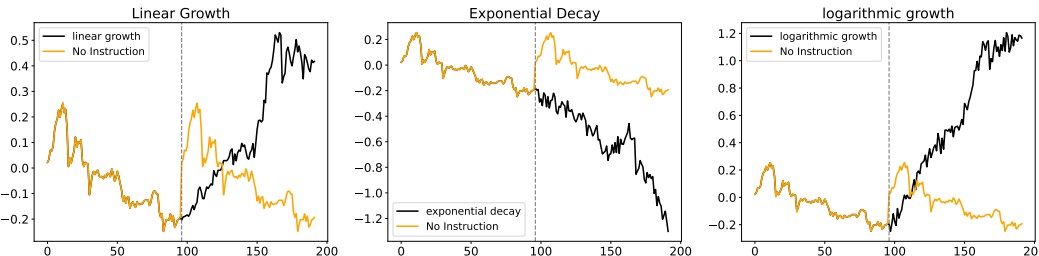

Figure 3: Comparison of different orders for the data generation: (a) Linear Growth, (b) Exponential Decay, and (c) logarithm growth. The vertical line indicates the start of the forecast.

## 4.2 PROPOSED ARCHITECTURE

Our proposed framework employs a two-stage approach to integrate time series data and textual information for predictive tasks. This multimodal architecture leverages both unsupervised and supervised learning paradigms to align time series data with corresponding semantic information derived from textual context, improving interpretability and predictive performance (see Figure 4 for an architectural overview).

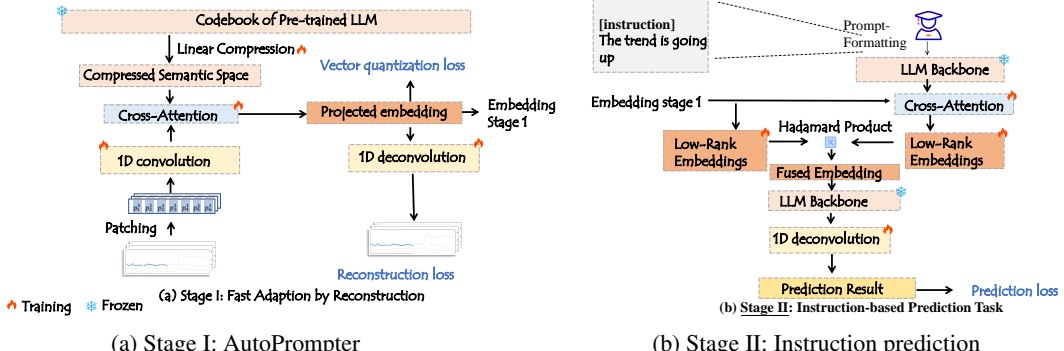

(a) Stage I: AutoPrompter  (b) Stage II: Instruction prediction

Figure 4: Model architecture

**Stage 1: AutoPrompter: Enhancing VQ-VAE for Unsupervised Alignment of Time Series and Pre-trained Language Codebook**

In the first stage, we propose a novel mechanism, AutoPrompter, which serves as a bridge, translating time series data into a compressed, semantically meaningful text embedding space. By introducing cross-attention and a pre-trained language codebook to the classical VQ-VAE (Vector Quantised-Variational AutoEncoder; (Van Den Oord et al., 2017)), AutoPrompter quantizes the time series space, allowing it to align effectively with text embeddings. This translation helps capture the underlying semantic patterns within the time series data, facilitating meaningful connections with textual information. The key components are as follows:

1. **Time Series Embedding**: The time series data, $\mathbf{X}$, is processed through a Convolutional Neural Network (CNN), producing a high-dimensional temporal embedding, $\mathbf{z}_{ts}$, that captures key temporal patterns.

2. **Text Embedding with Pre-trained Codebook and Linear Compression**: A pre-trained language model, equipped with a fixed vector quantization codebook, provides discrete text embeddings, $\mathbf{z}_{txt}$. These embeddings are subsequently passed through a linear compression layer to obtain a compressed semantic representation, $\mathbf{z}_{txt\_compressed}$. This compression reduces the dimensionality of the text embeddings, facilitating efficient alignment with the time series embeddings while preserving essential semantic information.

3. **Cross-Attention Alignment**: A cross-attention mechanism aligns the time series embedding $\mathbf{z}_{ts}$ with the compressed text embedding $\mathbf{z}_{txt\_compressed}$. In this setup, $\mathbf{z}_{ts}$ acts as the

query, and $\mathbf{z}_{\text{txt\_compressed}}$ serves as the key-value pair. This interaction facilitates the integration of temporal and semantic information, producing an aligned multimodal representation, $\mathbf{z}_{\text{aligned}}$.

4. **Self-Supervised Learning**: Our model is trained using self-supervised learning to align time series embeddings with meaningful text embeddings while preserving key characteristics of the original time series data. The training minimizes two key losses:

1. **Reconstruction Loss** ($\mathcal{L}_{\text{recon}}$) 2. **Vector Quantization Loss** ($\mathcal{L}_{\text{vq}}$)

- **Vector Quantization Loss** ($\mathcal{L}_{\text{vq}}$): The Vector Quantization Loss ensures that the encoder outputs are quantized by aligning them with the nearest vectors from a fixed, pre-trained language codebook. This encourages the model to represent time series data using a discrete set of embeddings from the codebook, facilitating alignment between the input data and text-based embeddings.

  Unlike traditional VQ-VAE, where the codebook is updated during training, our model employs a fixed codebook. However, because we apply a linear compression layer to the text embeddings, we use two distinct terms to maintain effective alignment:

  - **Quantization Latent Loss** ($\mathcal{L}_{\text{vq\_latent}}$) ensures that the quantized embeddings (obtained from the codebook) closely match the encoder's output.
  - **Commitment Loss** ($\mathcal{L}_{\text{commit}}$) encourages the encoder's output to remain close to the quantized embeddings. This ensures the model consistently uses the closest embedding in the codebook, preventing the encoder from drifting too far from the quantized space.

  These losses are defined as:

$$\mathcal{L}_{\text{vq\_latent}} = \|\mathbf{z}_{\text{aligned}} - \text{sg}(\mathbf{z}_{\text{encoder}})\|_2^2$$
$$\mathcal{L}_{\text{commit}} = \|\mathbf{z}_{\text{encoder}} - \text{sg}(\mathbf{z}_{\text{aligned}})\|_2^2$$
$$\mathcal{L}_{\text{vq}} = \mathcal{L}_{\text{vq\_latent}} + \mathcal{L}_{\text{commit}}$$

  where:

  - $\mathbf{z}_{\text{encoder}}$ is the output of the encoder.
  - $\mathbf{z}_{\text{aligned}}$ refers to the quantized embeddings obtained from the vector quantizer.
  - $\text{sg}(\cdot)$ is the stop-gradient operator, which ensures gradients don't flow back to the quantizer during optimization.

  By jointly minimizing these two components, we ensure that the encoder's outputs align well with the codebook, while keeping the representations compact and consistent.

- **Reconstruction Loss** ($\mathcal{L}_{\text{recon}}$): The Reconstruction Loss measures how well the reconstructed time series $\hat{\mathbf{x}}$, generated by the decoder, matches the original input $\mathbf{x}$. This loss ensures that essential information from the original time series is preserved:

$$\mathcal{L}_{\text{recon}} = \|\mathbf{x} - \hat{\mathbf{x}}\|_1$$

- **Total Loss**: The final loss function is a weighted sum of the Vector Quantization Loss and the Reconstruction Loss:

$$\mathcal{L}_{\text{total}} = \mathcal{L}_{\text{vq}} + \lambda_{\text{recon}} \cdot \mathcal{L}_{\text{recon}}$$

  where $\lambda_{\text{recon}}$ is a hyperparameter that balances the importance of the reconstruction task relative to the vector quantization. We further investigate the sensitivity of this parameter in Appendix H.

**Stage 2: Supervised Multimodal Fusion for Prediction** In this stage, we perform supervised time series prediction, conditioned on textual information. This leverages the embeddings learned in Stage 1, combined with a Large Language Model (LLM), to improve prediction accuracy.

1. **Text Embedding Extraction**: A pre-trained LLM extracts embeddings from textual inputs (e.g., "orders" or instructions). These embeddings capture the semantic context of the text, enriching the model's understanding for time series prediction.

2. **Multimodal Fusion**: Time series embedding works as query to get text embeddings combined using a cross-attention mechanism, which dynamically identifies relevant parts of both data sources. And then the embeddings are compressed to half of original dimension to further extract the essential information. This is followed by an adaptive Hadamard product, where time series embeddings $A$, text embeddings $B$, and a learnable weight matrix $W$ interact element-wise to form the joint representation:

$$C_{ij} = A_{ij} \cdot W_{ij} \cdot B_{ij}$$

3. **Supervised Training Objective**: The fused representation is processed by the LLM, whose output is passed into a CNN-based decoder (serving as the AutoPrompter decoder to translate from word embeddings to time-series again). The model is trained to minimize the mean absolute error (MAE) between predicted and actual values:

$$L_{\text{MAE}} = \frac{1}{N} \sum_{i=1}^{N} |y_i - \hat{y}_i|$$

**Key Innovation**: The core innovation of our framework is the semantic alignment of time series data with textual context through self-supervised learning. This approach enhances both interpretability and predictive accuracy by integrating multimodal data sources, making it particularly effective for applications where numerical signals and natural language descriptions are interdependent.

## 5 EXPERIMENTS

### 5.1 EXPERIMENTS SETUP

In this section, we present the experimental framework used to evaluate the performance of our proposed models, **AutoPrompter** and **TITSP**. The experiments are designed to assess key aspects such as unsupervised alignment, zero-shot generalization, and overall performance metrics.

For computational efficiency, we use **Qwen2-1.5B** (Yang et al., 2024) as our backbone language model, although other language models could also be used.

- **Stage 1**: To enhance both performance and resource efficiency, we employ the pre-trained codebook from Qwen, which is linearly compressed to $512$ dimensions. This dimensionality was chosen based on empirical evidence, as illustrated in Figure 20 in the Appendix, which demonstrates an optimal balance between these factors.

- **Stage 2**: The parameters from Stage 1 are kept fixed, and we perform end-to-end training using our designed loss function.

### 5.2 AUTOPROMPTER: UNSUPERVISED ALIGNMENT EVALUATION

We evaluate the **AutoPrompter** using key metrics: **reconstruction loss**, **zero-shot performance**, and **latent space visualization**. Our approach is compared against several baseline architectures to highlight the contributions of the model's components.

**Baselines**: The following baseline models are introduced for comparison:

- **No Cross-Attention**: This baseline excludes the cross-attention mechanism between time series and text embeddings. Instead, it directly employs a traditional Vector Quantized Variational Autoencoder (VQ-VAE) to map the time series embeddings to the nearest or most similar embeddings in the fixed pre-trained language codebook.

- **Reduced Codebook Size**: Employs a smaller compressed codebook to evaluate its impact on performance. The codebook is compressed into 50.

**Zero-shot Performance**: Table 1 summarizes the performance of AutoPrompter and also its zero-shot generalization performance when training on ETTh1 and testing on various datasets. The Reconstruction loss will demonstrate how well the information of time-series is represented using

| Dataset | AutoPrompter | AutoPrompter (Zero-shot) | No Cross-Attention | No Cross-Attention (zero-shot) | Reduced Codebook |
|---|---|---|---|---|---|
| ETTh1 | **0.018** | **0.018** | 0.104 | 0.104 | 1.025 |
| National Illness | **0.029** | *0.031* | 0.138 | 0.173 | 1.141 |
| Traffic | **0.027** | *0.056* | 0.273 | 0.475 | 2.131 |
| Exchange Rate | **0.005** | *0.015* | 0.820 | 1.090 | 1.053 |
| Weather | **0.007** | *0.008* | 0.540 | 0.950 | 1.101 |
| ETTm1 | **0.019** | *0.030* | 0.147 | 0.102 | 1.512 |
| ETTh2 | **0.011** | *0.015* | 0.129 | 0.145 | 1.357 |
| Electricity | **0.035** | *0.058* | 0.255 | 0.410 | 2.092 |
| ETTm2 | **0.019** | *0.026* | 0.122 | 0.192 | 0.947 |
| Lorenz Time | **0.010** | *0.011* | 0.132 | 0.212 | 0.133 |

Table 1: Zero-shot reconstruction loss comparison across different architectures. The best-performing value for each dataset is in **bold**, and the second-best is in *italics*.

compressed language codebook. The results clearly demonstrate that the AutoPrompter achieves superior performance across all datasets.

**Latent Space Visualization (t-SNE)**: To examine the alignment of time series and text embeddings, we visualize the latent space using t-SNE. Figure 5 shows that AutoPrompter with the compressed codebook results in well-clustered embeddings. In comparison, a trainable randomly initialized codebook produces scattered, unstructured embeddings after training.

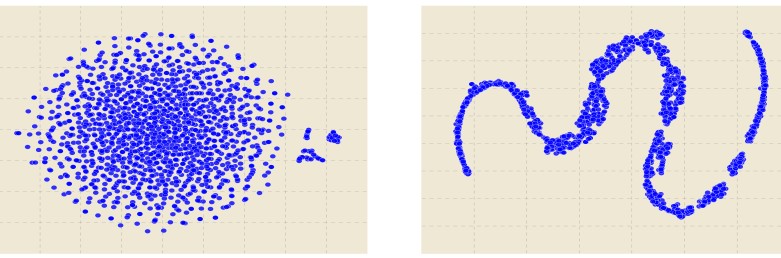

Figure 5: t-SNE Visualization: with pre-trained Codebook (Right) vs. with trainable Randomly initialized Codebook (Left)

**Conclusion**: The evaluation results demonstrate that the **AutoPrompter** significantly outperforms the baseline models in zero-shot tasks. The t-SNE visualization emphasizes the importance of the codebook in generating semantically structured embeddings, while the codebook size analysis suggests that an optimal size enhances performance, underscoring the critical role of the codebook in our framework.

## 5.3 OVERALL PERFORMANCE EVALUATION OF TITSP

To provide a thorough assessment of the TITSP model's performance, we focus on three critical metrics: **Order Compliance Rate**, **Zero-shot Ability**, and the performance of TITSP for long sequence for the instruction.

- **Order Compliance Rate**: To provide a thorough assessment of the TITSP model's performance, we introduce the *Compliance Rate* metric to quantitatively evaluate how well the model adheres to specified actions guiding time series predictions. This metric reflects the proportion of time steps where the model's predictions align with the expected trends or patterns dictated by the actions.

  The Compliance Rate $R$ is defined as: $R = \sum_{t=1}^{T_1} C_t$, where $C_t$ is an indicator variable such that $C_t = 1$ if the prediction at time step $t$ adheres to the prescribed action, and $C_t = 0$ otherwise. Here, $T_1$ represents the total number of time steps considered.

  A higher compliance rate indicates better adherence to specified actions, showcasing the model's ability to incorporate desired behaviors into its predictions. For detailed definitions and calculations, refer to Appendix E. In this study, we compare our method with Time-LLM (Jin et al., 2023), which uses natural language as input with well-designed prompts

to aid predictions. Additionally, we introduce two other baselines that use only LLMs with a prompt concatenating both instructions and time series, as well as UniTime (Liu et al., 2024b) and GPT4MTS(Jia et al., 2024). We adapt GPT4MTS to Qwen4MTS with the same language model we use so that we can fairly compare. Further insights into the Time-LLM prompt are provided in Appendix A. We also provide the detail implementation of Qwen4MTS and UniTime(Qwen) in Appendix I. Pure LLM results are also provided with Llama-3.1-8B-Instruct (Dubey et al., 2024), the desinged prompts are provided in Appendix I. We did not compare with Chronos (Ansari et al., 2024) and UniTS (Gao et al., 2024) as they only input time-series.

| Instruction | TITSP | | Time-LLM | | Qwen4MTS | | UniTime (Qwen) | | Llama-3.1-8B | |
|---|---|---|---|---|---|---|---|---|---|---|
| Metric | CR | MSE | CR | MSE | CR | MSE | CR | MSE | CR | MSE |
| Linear Growth and Linear Decay | **0.83** | **1.15** | 0.38 | 3.45 | 0.69 | 1.90 | 0.54 | 2.73 | 0.32 | 4.95 |
| Linear Growth and Linear Decay | **0.79** | **1.17** | 0.49 | 2.85 | **0.79** | 1.34 | 0.57 | 2.28 | 0.41 | 2.80 |
| Linear Trend Up | 0.90 | **1.03** | 0.63 | 1.71 | 0.76 | 1.08 | 0.63 | 1.65 | **0.91** | 1.15 |
| Linear Trend Down | **0.87** | **0.88** | 0.64 | 1.55 | 0.71 | 1.36 | 0.51 | 1.59 | 0.85 | 0.92 |
| Exponential Growth | **0.89** | **1.33** | 0.58 | 2.59 | 0.63 | 2.07 | 0.60 | 2.38 | 0.58 | 2.35 |
| Exponential Decay | **0.84** | **1.25** | 0.56 | 2.26 | 0.67 | 2.10 | 0.69 | 2.05 | 0.46 | 2.39 |
| Keep Stable | **0.98** | 0.35 | 0.76 | 0.76 | 0.93 | 0.48 | 0.83 | 0.62 | 0.95 | **0.33** |
| Decrease Amplitude | **0.90** | 0.91 | 0.85 | 1.04 | **0.90** | **0.84** | 0.79 | 1.09 | 0.52 | 1.89 |
| Increase Amplitude | **0.94** | **0.94** | 0.79 | 1.20 | 0.89 | 0.96 | 0.81 | 1.03 | 0.75 | 1.35 |
| Logarithmic Growth | 0.77 | 1.65 | 0.49 | 2.31 | **0.79** | **1.55** | 0.60 | 1.73 | 0.55 | 1.94 |
| Logarithmic Decay | **0.83** | **1.68** | 0.48 | 2.19 | 0.81 | 1.69 | 0.67 | 2.04 | 0.63 | 2.60 |

Table 2: Comparison of Compliance Rate (CR) and MSE for TITSP, Time-LLM, Qwen4MTS, UniTime, and Llama-3.1-8B across various instructed actions with highlighted best (in **bold**) and second-best (underline) results.

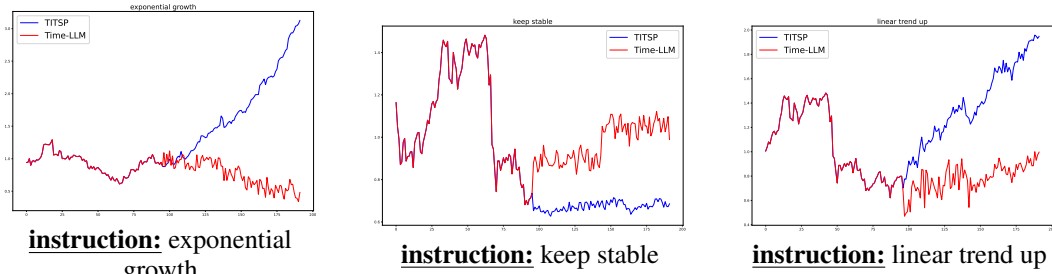

**instruction:** exponential growth     **instruction:** keep stable     **instruction:** linear trend up

Figure 6: comparison of Time-LLM and TITSP. Our method succeed in learning the dependency of prediction and instruction, (red: Time-LLM, blue: TITSP)

- **Zero-shot Ability**: The TITSP model demonstrates strong zero-shot generalization, effectively adapting to variations of action instructions without retraining. Across a range of similar expressions, it achieves high compliance rates, showcasing its robust ability to understand and follow instructions with minimal performance degradation. The detailed performance for selected instructions is presented in Table 3 in Appendix, where the model maintains near-perfect compliance, indicating superior adaptability across different contexts.

To further highlight the TITSP model's zero-shot generalization ability, we present time series predictions for various instructions. These examples in Figure 7 demonstrate the model's capacity to adapt to new, unseen patterns, reinforcing its robustness in generalizing across different instructions.

| Training Instruction | Test Instruction | Compliance Rate | MSE |
|---|---|---|---|
| *Linear Trend Up* | *Linear Upward* | 0.81 | 1.27 |
| | *Linear Goes Up* | 0.89 | 0.93 |
| | *Linear Growth* | 0.80 | 1.13 |
| *Linear Growth and Decay* | *Linear Up and Down* | 0.71 | 1.93 |
| *Linear Decay and Growth* | *Linear Down and Up* | 0.73 | 1.51 |
| *Exponential Growth* | *Exponential Upward* | 0.71 | 1.93 |
| | *Exponential Goes Up* | 0.82 | 1.53 |

Table 3: Zero-shot performance of TITSP on selected test instructions.

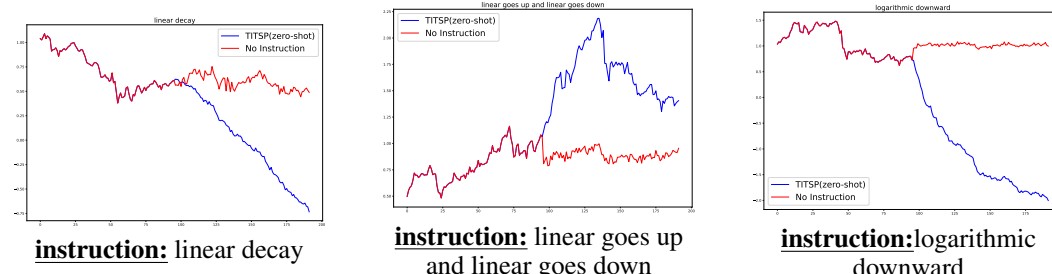

**instruction:** linear decay      **instruction:** linear goes up and linear goes down      **instruction:** logarithmic downward

Figure 7: Samples of Zero-shot about TITSP, which illustrate that TITSP has strong zero-shot ability, which is also an advantage of LLMs.

- **Keyword Extraction Capability**: The TITSP model exhibits remarkable performance in extracting relevant keywords from long sequences. The attention mechanism effectively identifies critical components in the input data, enhancing the model's ability to discern patterns and relationships in the time series. For visual insights, refer to the attention matrix provided in Appendix G.

**Conclusion:** The results underscore the efficacy of the TITSP model, particularly in its order compliance, zero-shot learning ability, and adeptness at keyword extraction. These strengths position TITSP as a powerful tool for time series prediction, capable of seamlessly integrating and adhering to textual instructions while maintaining high accuracy and interpretability.

To conclude this experimental section, we address the ongoing debate regarding the relevance of large language models (LLMs) in time-series prediction. While some researchers question their importance (Tan et al., 2024), others highlight their potential in sequence modeling (Liu et al., 2024a). Our work contributes to this discussion by systematically replacing LLMs with traditional architectures, such as transformers and multilayer perceptrons (MLPs), and evaluating the resulting performance as seen in Appendix D. Additionally, we conduct extensive ablation studies, detailed in Appendix D, to assess the significance of specific model components..

## 6 CONCLUDING REMARKS

In this paper, we introduced *Text-Informed Time Series Prediction* (TITSP), a novel framework that enhances time series forecasting by integrating domain-specific textual information. Extensive experiments across diverse datasets demonstrate that TITSP significantly outperforms traditional and existing multimodal approaches, improving both predictive accuracy and interpretability. Notably, TITSP exhibits robust zero-shot generalization, enabling effective deployment across various domains without extensive retraining. Our findings underscore the potential of multimodal methodologies to transform time series modeling, offering more accurate and versatile solutions for real-world applications. By bridging numerical data and contextual textual information, TITSP promises substantial impacts in fields requiring precise forecasting and decision-making.

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

**Roadmap.** This appendix provides additional experiments and details omitted from the main paper for conciseness. It is organized as follows:

- Section A: Explores the main disadvantage of Time LLM in handling textual context, motivating the present work.
- Section B: Complements the main paper's experiments with additional visualizations of TITSP's performance and evidence of its zero-shot capability.
- Section C: Details the data generation process, emphasizing base time series and transformations to incorporate context and text.
- Section D: Conducts extensive ablation studies to validate the rationale of each architectural element.
- Section E: Provides a formal definition of the compliance rate with descriptive examples.
- Section F: Presents evidence on Stage 1 and its reconstruction ability.
- Sections G and H: Explain how the model handles longer sequences and establish sensitivity to loss hyperparameters.

# TABLE OF CONTENTS

Table 4: Performance of Time-LLM under various textual modifications

| Modification | MSE | MAE |
|---|---|---|
| No change | 0.383 | 0.402 |
| Incorrect dataset description | 0.386 | 0.404 |
| Random mean, max, min values | 0.383 | 0.403 |
| Opposite trend description | 0.383 | 0.403 |

## A    LIMITATIONS OF TIME-LLM: INFLUENCE OF TEXTUAL PROMPTS

In this section, we investigate the limitations of Time-LLM by examining the impact of textual prompts on its performance. We systematically alter the dataset descriptions to assess whether changes in textual context, unrelated to the time series data, affect the model's predictions. The modifications tested include:

- Providing an incorrect dataset description.

- Assigning random mean, max, and min values between 0 and 1.

- Providing a description with an opposite trend.

To evaluate the model, we use a sequence length of 96 for both input and prediction on the ETTh1 dataset. The results are summarized in Table 4.

These results indicate that altering the textual descriptions had negligible effects on the performance of Time-LLM. The MSE and MAE values remained largely unchanged, suggesting that the model's predictions are primarily driven by the time series data, with minimal influence from the textual context. This observation underscores the limitations of using prompts in this particular setup and motivates the need for further exploration to determine the conditions under which text can meaningfully guide time-series predictions.

## B    OVERALL PERFORMANCE OF TITSP

This section presents the overall performance of the Text-Informed Time Series Prediction (TITSP) model, including visualizations of instruction-based time-series predictions and zero-shot samples.

### B.1    VISUALIZATION OF INSTRUCTION-BASED TIME-SERIES PREDICTIONS

In this subsection, we provide visualizations of the prediction results with and without instructions. These visualizations demonstrate the model's ability to adapt to various textual instructions, showcasing its versatility and accuracy.

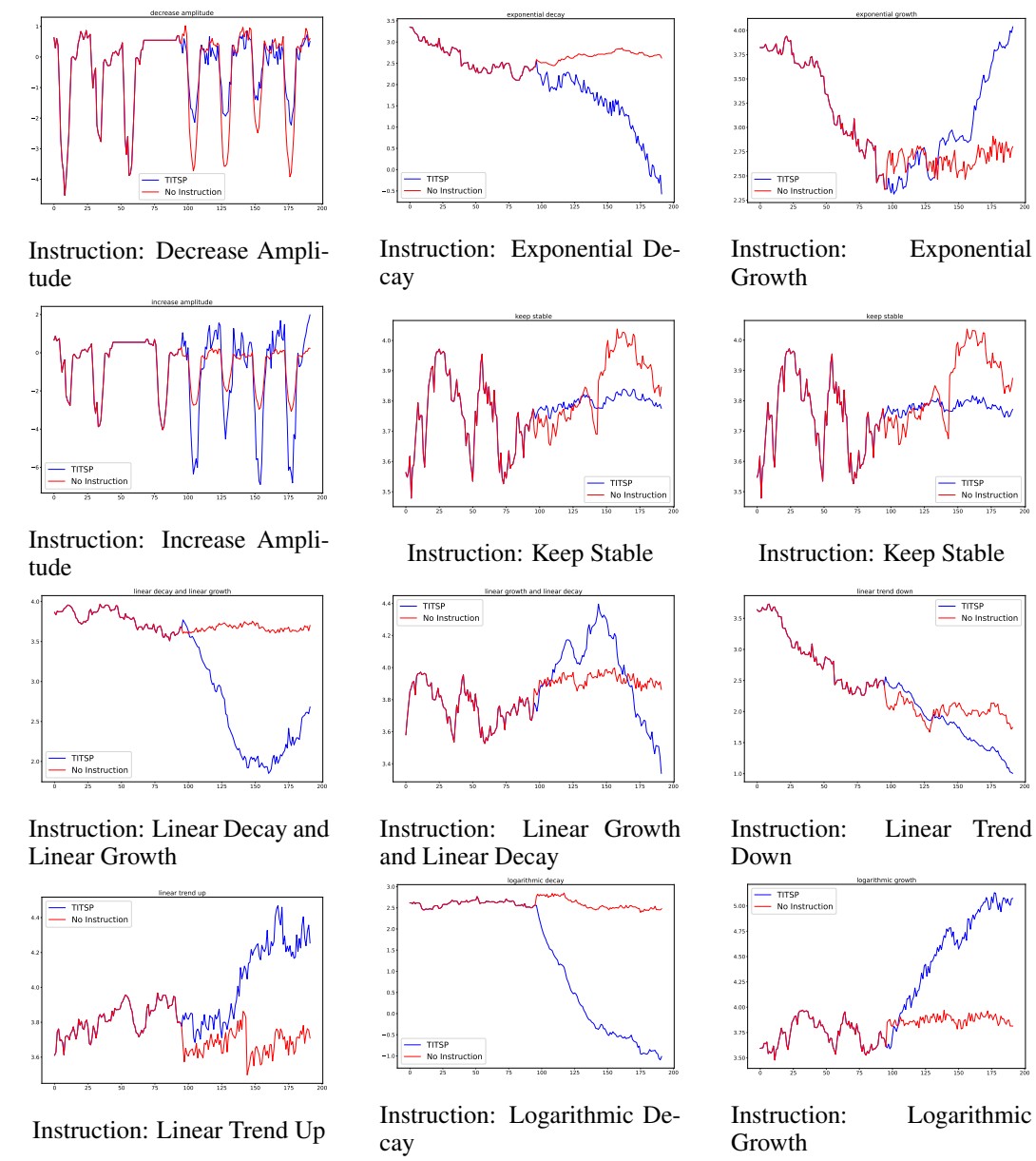

Instruction: Decrease Amplitude

Instruction: Exponential Decay

Instruction: Exponential Growth

Instruction: Increase Amplitude

Instruction: Keep Stable

Instruction: Keep Stable

Instruction: Linear Decay and Linear Growth

Instruction: Linear Growth and Linear Decay

Instruction: Linear Trend Down

Instruction: Linear Trend Up

Instruction: Logarithmic Decay

Instruction: Logarithmic Growth

Figure 8: Samples of prediction results with and without instructions

## B.2 ZERO-SHOT SAMPLES OF TITSP

In this subsection, we present zero-shot samples of the TITSP model. These samples demonstrate the model's ability to generalize to new, unseen instructions without additional training, highlighting its robustness and adaptability.

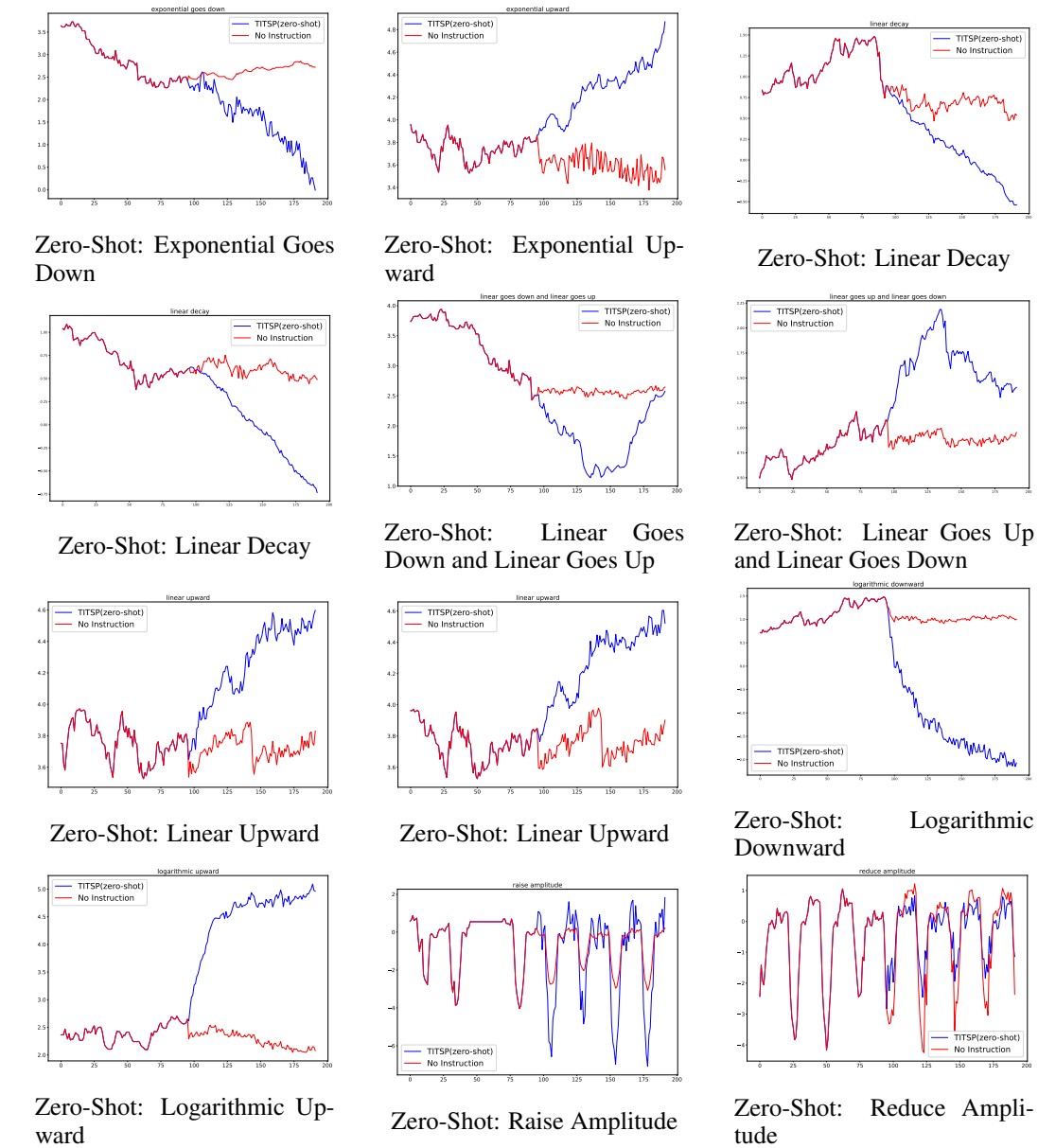

Figure 9: Samples of zero-shot prediction results

These visualizations and zero-shot samples provide a comprehensive overview of the TITSP model's capabilities, demonstrating its effectiveness in integrating textual instructions with time-series data.

### B.3 VISUALIZATION OF NORMAL PROMPTED TIME-SERIES PREDICTION

In this subsection, we present the full results of the TITSP model using only dataset descriptions and task descriptions.

We give here two examples of prompt that are used to show that they may differ from one dataset to another.

- "Dataset description: The Electricity Transformer Temperature (ETT) is a crucial indicator in the electric power long-term deployment. This dataset consists of 2 years data from two separated counties in China. To explore the granularity on the Long sequence

*time-series forecasting (LSTF) problem, different subsets are created, ETTh1, ETTh2 for 1-hour-level and ETTm1 for 15-minutes-level. Each data point consists of the target value "oil temperature" and 6 power load features. The train/val/test is 12/4/4 months. Task description: forecast the next 96 steps given the previous 96 steps information."*

- *"Traffic is a collection of hourly data from California Department of Transportation, which describes the road occupancy rates measured by different sensors on San Francisco Bay area freeways. Task description: forecast the next 96 steps given the previous 96 steps information."*

To ensure a fair comparison, we adapt GPT4TS and Time-LLM to QWEN2-1.5B (Yang et al., 2024). The input and output sequence length is set to 96. The results demonstrate that TITSP excels at building the dependency between prompt words and time-series input, fully utilizing the capabilities of large language models.

| Models | TITSP | | GPT4TS | | Time-LLM | | itransformer | | Rlinear | | PatchTST | |
|---|---|---|---|---|---|---|---|---|---|---|---|---|
| Metric | MSE | MAE | MSE | MAE | MSE | MAE | MSE | MAE | MSE | MAE | MSE | MAE |
| ETTh1 | 0.416 | 0.422 | 0.428 | 0.426 | 0.450 | 0.445 | 0.454 | 0.448 | 0.446 | 0.434 | 0.469 | 0.455 |
| ETTh2 | 0.406 | 0.409 | 0.405 | 0.413 | 0.427 | 0.430 | 0.432 | 0.434 | 0.422 | 0.420 | 0.441 | 0.437 |
| ETTm1 | 0.375 | 0.399 | 0.390 | 0.405 | 0.408 | 0.419 | 0.417 | 0.425 | 0.406 | 0.411 | 0.423 | 0.426 |
| ETTm2 | 0.365 | 0.389 | 0.373 | 0.399 | 0.395 | 0.414 | 0.402 | 0.418 | 0.390 | 0.406 | 0.404 | 0.418 |
| Weather | 0.342 | 0.382 | 0.354 | 0.395 | 0.382 | 0.405 | 0.383 | 0.407 | 0.375 | 0.399 | 0.387 | 0.407 |
| Traffic | 0.360 | 0.383 | 0.357 | 0.396 | 0.398 | 0.414 | 0.393 | 0.412 | 0.391 | 0.408 | 0.394 | 0.412 |
| Electricity | 0.357 | 0.381 | 0.351 | 0.392 | 0.402 | 0.415 | 0.392 | 0.409 | 0.396 | 0.409 | 0.389 | 0.408 |
| Exchange rate | 0.365 | 0.397 | 0.349 | 0.389 | 0.400 | 0.410 | 0.391 | 0.406 | 0.398 | 0.406 | 0.382 | 0.402 |
| 1st count | 11 | | 5 | | 0 | | 0 | | 0 | | 0 | |

Table 5: Comparison of models based on MSE and MAE metrics with highlighted best (red) and second-best (green) results

These results highlight the superior performance of TITSP in building dependencies between prompt words and time-series input, effectively utilizing the capabilities of large language models. The table presents a comprehensive comparison of various models, with the best and second-best results highlighted in red and green, respectively. This demonstrates the robustness and effectiveness of the TITSP model in time-series prediction tasks.

## C  DATA GENERATION

In this section, we provide a detailed definition of the data generation process to ensure high-quality datasets for evaluation. We generate data from various datasets, including ETTh1, ETTh2, ETTm1, ETTm2, weather, traffic, electricity, exchange_rate.

In this table:

- `np.arange(x)` generates a sequence of time steps.

- *time* represents the time variable in the time series.

- *transition time* is set to 0.5 for uniform experiments.

- $A$, $B$, and $C$ are constants defining the amplitude and the slopes. In our case, we set $A \sim \mathcal{U}(0, 0.5)$, $B$ and $C \sim \mathcal{U}(0.01, 0.015)$.

The detailed equations for *Linear Trend Up*, *Linear Trend Down*, *Linear Growth and Linear Decay*, and *Linear Decay and Linear Growth* are listed as follows:

$$\begin{cases} \text{Original slope } m = \text{linear regression(time, batch\_y\_i)} \\ \text{New slope } m' = -m + \delta, \quad \delta \sim \mathcal{U}(0.01, 0.015) \\ \text{New sequence } y' = \text{batch\_y\_i} + m' \times \text{time} \end{cases} \quad (4)$$

| Action | Description | Mathematic Function | Generated Dataset |
|---|---|---|---|
| Linear Trend Up | Linear increase over time | see Equation 4 | weather, exchange_rate |
| Linear Trend Down | Linear decrease over time | see Equation 5 | weather, exchange_rate |
| Exponential Growth | Exponential increase over time | prediction $\times \exp(B \times$ np.arange$(x))$ | weather, exchange_rate, electricity |
| Exponential Decay | Exponential decrease over time | prediction $\times$ $\exp(-B$ $\times$ np.arange$(x))$ | weather, exchange_rate, electricity |
| Logarithmic Growth | Logarithmic growth over time | prediction $+$ $C$ $\times$ $\log(1+$np.arange$(x))$ | weather, exchange_rate, electricity |
| Logarithmic Decay | Logarithmic decay over time | prediction $-$ $C$ $\times$ $\log(1+$np.arange$(x))$ | weather, exchange_rate, electricity |
| Keep Stable | Constant value | last input point | All |
| Linear Growth and Linear Decay | Linear increase followed by decrease | see Equation 6 | weather, exchange_rate |
| Linear Decay and Linear Growth | Linear decrease followed by increase | see Equation 7 | weather, exchange_rate |
| Increase Amplitude | Scale up predictions by a factor | prediction $\times (1+A)$ | ETTh1, ETTh2, ETTm1, ETTm2, traffic |
| Decrease Amplitude | Scale down predictions by a factor | prediction $\times (1-A)$ | ETTh1, ETTh2, ETTm1, ETTm2, traffic |

Table 6: Summary of actions, their descriptions, corresponding mathematical functions, and generated datasets.

$$\begin{cases} \text{Original slope } m = \text{linear regression(time, batch\_y\_i)} \\ \text{New slope } m' = -m - \delta, \quad \delta \sim \mathcal{U}(0.01, 0.015) \\ \text{New sequence } y' = \text{batch\_y\_i} + m' \times \text{time} \end{cases} \quad (5)$$

$$\text{trend} = \begin{cases} \text{initial\_slope} \times \text{time}(t), & t < \text{transition\_time} \\ \text{initial\_slope} \times \text{time(transition\_time)} - \text{decline\_slope} \times (t - \text{transition\_time}), & t \geq \text{transition\_time} \end{cases} \quad (6)$$

$$\text{trend} = \begin{cases} -\text{initial\_slope} \times \text{time}(t), & t < \text{transition\_time} \\ -\text{initial\_slope} \times \text{time(transition\_time)} + \text{increase\_slope} \times (t - \text{transition\_time}), & t \geq \text{transition\_time} \end{cases} \quad (7)$$

For this part, we provide some examples of data generation:

### C.0.1 LORENZ TIME SERIES

In addition to the generation of synthetic datasets for general time series tasks, we also generate data based on the Lorenz system, which is commonly used to model chaotic dynamics. The Lorenz system is governed by the following set of ordinary differential equations (ODEs):

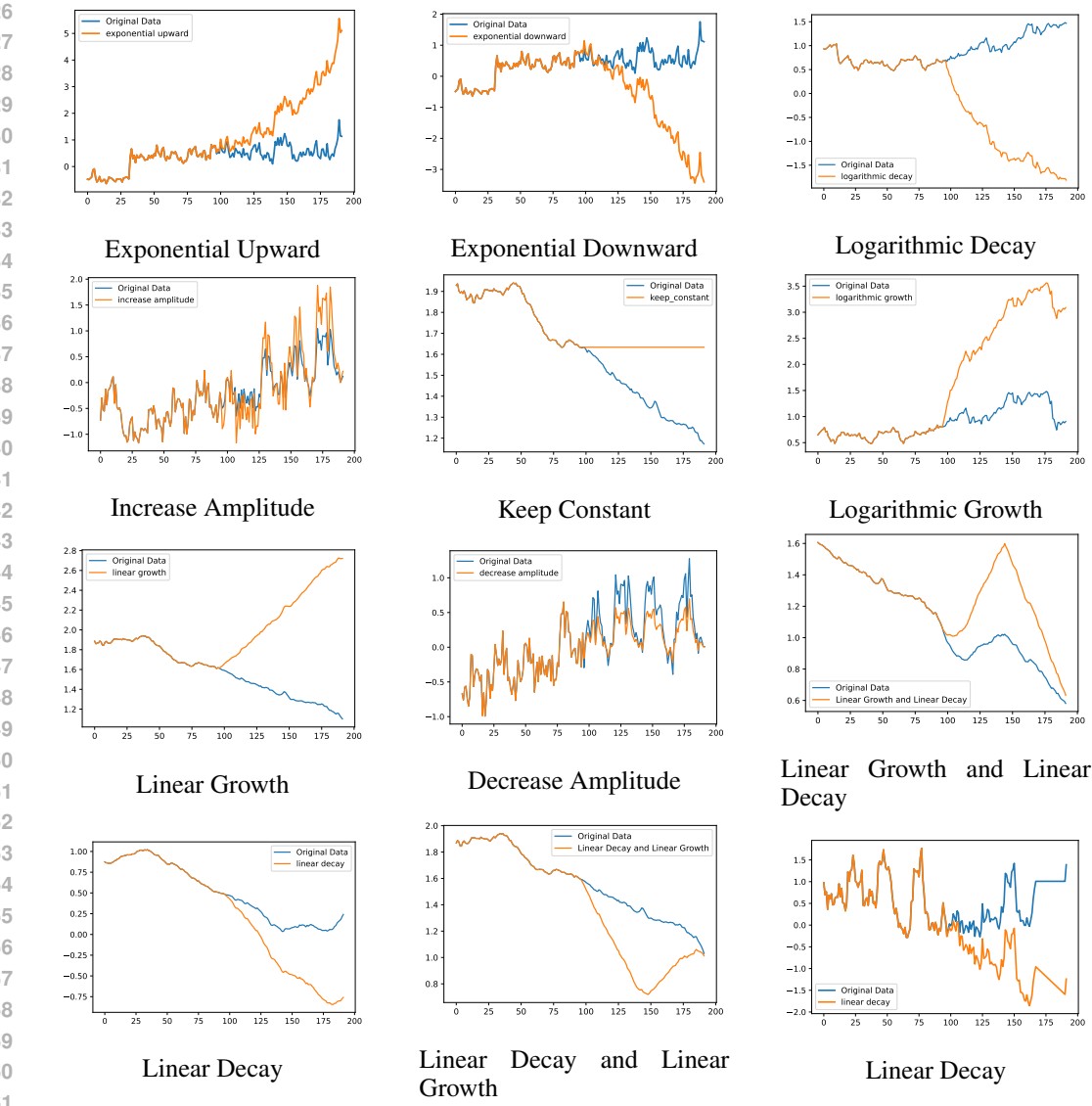

Figure 10: Examples of data generation

$$\frac{dx}{dt} = \sigma(y - x) \tag{8}$$

$$\frac{dy}{dt} = x(\rho - z) - y \tag{9}$$

$$\frac{dz}{dt} = xy - \beta z \tag{10}$$

where $x$, $y$, and $z$ represent the state variables, and $\sigma$, $\rho$, and $\beta$ are system parameters. For the generation of Lorenz time series data, we typically set the parameters as follows:

- $\sigma = 10$

- $\rho = 28$

- $\beta = \frac{8}{3}$

To numerically solve this system, we use methods such as the fourth-order Runge-Kutta method, initializing the system with a set of initial conditions $(x_0, y_0, z_0)$. The time series for each of the state variables ($x(t)$, $y(t)$, and $z(t)$) exhibits chaotic behavior, characterized by high sensitivity to the initial conditions.

The generated Lorenz time series data provide a rich dataset for testing models on chaotic systems, as it challenges the model's ability to predict complex, non-linear temporal dependencies. The chaotic nature of the Lorenz system makes it an excellent benchmark for evaluating time series forecasting models.

# D ABLATION STUDY

In this section, we conduct an ablation study to understand the importance of various components in the TITSP model. Specifically, we investigate the significance of the weight matrix $W$ and the role of Large Language Models (LLMs) in time-series prediction.

## D.1 IMPORTANCE OF $W$

To evaluate the importance of the weight matrix $W$, we directly use the following equation:

$$C_{ij} = A_{ij} \cdot B_{ij}$$

and compare the performance with the baseline model. The results are presented in Section D.2.

## D.2 ARE LLMS REALLY IMPORTANT IN TIME-SERIES PREDICTION?

In this part, we follow the settings of (Tan et al., 2024) to replace LLMs with a Multi-Layer Perceptron (MLP) and compare the performance. The results are summarized in Table 7.

| Action | TITSP | | TITSP (without $W$) | | TITSP (without LLM, with MLP) | | Time-LLM (Jin et al., 2023) | |
|---|---|---|---|---|---|---|---|---|
| Metric | Compliance Rate | MSE | Compliance Rate | MSE | Compliance Rate | MSE | Compliance Rate | MSE |
| Linear Trend Up | 0.90 | 1.03 | 0.79 | 1.19 | 0.59 | 2.83 | 0.63 | 1.71 |
| Linear Trend Down | 0.87 | 0.88 | 0.80 | 0.89 | 0.60 | 2.23 | 0.64 | 1.55 |
| Exponential Growth | 0.89 | 1.33 | 0.81 | 1.57 | 0.52 | 3.19 | 0.58 | 2.59 |
| Exponential Decay | 0.84 | 1.25 | 0.79 | 1.42 | 0.44 | 3.25 | 0.56 | 2.26 |
| Keep Stable | 0.98 | 0.35 | 0.82 | 0.73 | 0.79 | 0.70 | 0.83 | 0.76 |
| Decrease Amplitude | 0.90 | 0.91 | 0.79 | 1.30 | 0.67 | 1.51 | 0.85 | 1.04 |
| Increase Amplitude | 0.94 | 0.94 | 0.68 | 2.05 | 0.59 | 2.02 | 0.79 | 1.20 |
| Logarithmic Growth | 0.77 | 1.65 | 0.69 | 1.69 | 0.53 | 2.58 | 0.49 | 2.31 |
| Logarithmic Decay | 0.83 | 1.68 | 0.71 | 1.74 | 0.54 | 2.36 | 0.48 | 2.19 |
| Linear Growth and Linear Decay 1 | 0.83 | 1.15 | 0.63 | 1.89 | 0.54 | 3.09 | 0.38 | 3.45 |
| Linear Growth and Linear Decay 2 | 0.79 | 1.17 | 0.65 | 1.97 | 0.64 | 2.28 | 0.49 | 2.85 |

Table 7: Comparison of Compliance Rate and MSE for TITSP (with various configurations) and Time-LLM across various instructed actions.

In summary, both LLMs and the weight matrix $W$ play important roles in the task of time-series prediction. Specifically, $W$ helps in dynamically building the complex dependency of input instructions and giving weights to the input embedding. LLMs play a crucial role in modeling complex patterns, which are essential for accurate time-series prediction.

# E COMPLIANCE RATE DEFINITION

In time-series prediction tasks that are guided by specific actions or directives (as outlined in Table 6), it is essential to quantitatively evaluate how well the model adheres to these prescribed behaviors. We introduce the **Compliance Rate** as a metric that measures the proportion of cases where the model's predictions align with the expected trends or patterns dictated by the actions.

## E.1 NOTATION

We define the following terms:

- $\hat{y}_t$: The model's predicted value at time $t$ *with* the specified action.

- $y_t$: The model's predicted value at time $t$ *without* the specified action (i.e., the baseline prediction).

- $T$: The total number of time steps in the time series.

- $\mathbf{1}\{\cdot\}$: The indicator function, which equals 1 if the condition inside is true, and 0 otherwise.

- $\delta$: A small positive threshold indicating the minimum acceptable rate of change (slope).

- $\epsilon$: A small threshold value representing acceptable fluctuation for the "Keep Stable" action.

- $t_{\text{transition}}$: The transition time point for actions involving a change in trend.

### E.2 GENERAL DEFINITION

The compliance rate $C$ is computed as:

$$C = \frac{\sum_{i=1}^{N} \mathbf{1}\{\text{Compliance Condition for series } i \text{ is satisfied}\}}{N} \tag{11}$$

where $N$ is the total number of time series (or segments) being evaluated.

### E.3 COMPLIANCE CONDITIONS PER ACTION

The compliance conditions evaluate the *overall trends* of the model's predictions using statistical methods such as linear regression, which accounts for inherent fluctuations in the data.

1. **Linear Trend Up**:
   - **Model Fitting**: Fit the following models to $\hat{y}_t$:
     (a) **Linear Model**: $\hat{y}_t = mt + c$
     (b) **Exponential Model**: $\hat{y}_t = ae^{bt}$
     (c) **Logarithmic Model**: $\hat{y}_t = a\ln(t) + c$
   - **Goodness-of-Fit**: Compute $R^2$ for each model.
   - **Compliance Condition**: The linear model has the highest $R^2$ *and* the slope $m$ satisfies:
     $$m \geq \delta \tag{12}$$

2. **Linear Trend Down**: Similar to Linear Trend Up, but the compliance condition is $m \leq -\delta$.

3. **Exponential Growth**:
   - **Model Fitting**: Fit the following models to $\hat{y}_t$:
     (a) **Exponential Model**: $\hat{y}_t = ae^{bt}$
     (b) **Linear Model**: $\hat{y}_t = mt + c$
     (c) **Logarithmic Model**: $\hat{y}_t = a\ln(t) + c$
   - **Goodness-of-Fit**: Compute $R^2$ for each model.
   - **Compliance Condition**: The exponential model has the highest $R^2$ *and* the rate parameter $b$ satisfies:
     $$b \geq \delta \tag{13}$$

4. **Exponential Decay**: Similar to Exponential Growth, but the compliance condition is $b \leq -\delta$.

5. **Logarithmic Growth**:
   - **Model Fitting**: Fit the following models to $\hat{y}_t$:
     (a) **Logarithmic Model**: $\hat{y}_t = a\ln(t) + c$
     (b) **Linear Model**: $\hat{y}_t = mt + c$
     (c) **Exponential Model**: $\hat{y}_t = ae^{bt}$
   - **Goodness-of-Fit**: Compute $R^2$ for each model.

- **Compliance Condition**: The logarithmic model has the highest $R^2$ *and* the coefficient $a$ satisfies:

$$a \geq \delta \tag{14}$$

6. **Logarithmic Decay**: Similar to Logarithmic Growth, but the compliance condition is $a \leq -\delta$.

7. **Keep Stable**: The compliance condition is:

$$\sigma \leq \epsilon \tag{15}$$

where $\sigma$ is the standard deviation of $\hat{y}_t$.

8. **Linear Growth and Linear Decay**:

- **Segment Division**: Divide the time series into two segments at $t_{\text{transition}}$:
    - First segment: $t \in [1, t_{\text{transition}} - 1]$
    - Second segment: $t \in [t_{\text{transition}}, T]$
- **Model Fitting for Each Segment**: Fit linear, exponential, and logarithmic models to each segment.
- **Compliance Condition**: The linear model has the highest $R^2$ in both segments, with slopes satisfying:

$$\begin{cases} m_1 \geq \delta & \text{(First Segment)} \\ m_2 \leq -\delta & \text{(Second Segment)} \end{cases} \tag{16}$$

9. **Linear Decay and Linear Growth**: Same as above, but with conditions:

$$\begin{cases} m_1 \leq -\delta & \text{(First Segment)} \\ m_2 \geq \delta & \text{(Second Segment)} \end{cases} \tag{17}$$

10. **Increase Amplitude**:

- **Compliance Condition** (Evaluated per time step): For each time step $t$:

$$|\hat{y}_t - y_t \times (1 + A)| \leq \epsilon \tag{18}$$

11. **Decrease Amplitude**:

- **Compliance Condition** (Evaluated per time step): For each time step $t$:

$$|\hat{y}_t - y_t \times (1 - A)| \leq \epsilon \tag{19}$$

### E.4 CALCULATING THE COMPLIANCE RATE

The compliance rate $C$ is calculated based on the action:

1. For actions evaluated per action:

    (a) Evaluate the compliance condition for each series.

    (b) Aggregate and normalize the results:

$$C = \left( \frac{\text{Number of Series Satisfying Compliance}}{N} \right) \times 100\% \tag{20}$$

2. For actions evaluated per time step:

    (a) Evaluate compliance at each time step.

    (b) Aggregate and normalize across all time steps:

$$C = \left( \frac{\text{Number of Time Steps Satisfying Compliance}}{N_{\text{total}}} \right) \times 100\% \tag{21}$$

where $N_{\text{total}} = N \times T$.

### E.5   EXAMPLE CALCULATION

Suppose we evaluate 100 time series with the **Linear Trend Up** action. If 85 of these series have a slope $m$ such that $m \geq \delta$, the compliance rate is:

$$C = \left(\frac{85}{100}\right) \times 100\% = 85\% \tag{22}$$

### E.6   DISCUSSION

- **Threshold $\delta$**: The threshold $\delta$ is chosen based on domain knowledge or acceptable performance criteria. In our case, we set $\delta = 0.01$ based on the acceptable slope range of (0.01, 0.015).

- **Threshold $\epsilon$**: The threshold $\epsilon$ for "Keep Stable" actions is set at 0.001.

## F   RECONSTRUCTION ABILITY OF AUTOPROMPTER

### F.1   ZERO-SHOT ABILITY

In this section, we present the zero-shot generalization performance of the proposed AutoPrompter method. The model's ability to follow different trend instructions is measured in terms of the **Compliance Rate** and **Mean Squared Error (MSE)**. Table 3 provides a detailed performance matrix, showing how well the model adheres to various trends without any fine-tuning.

| Performance Matrix | | Compliance Rate | MSE |
|---|---|---|---|
| **Linear Trend Up** | Linear Upward | 0.81 | 1.27 |
| | Linear Goes Up | 0.89 | 0.93 |
| | Linear Growth | 0.80 | 1.13 |
| **Linear Trend Down** | Linear Downward | 0.83 | 1.03 |
| | Linear Goes Down | 0.88 | 0.83 |
| | Linear Decay | 0.74 | 1.63 |
| **Exponential Growth** | Exponential Upward | 0.71 | 1.93 |
| | Exponential Goes Up | 0.82 | 1.53 |
| **Exponential Decay** | Exponential Downward | 0.80 | 1.43 |
| | Exponential Goes Down | 0.73 | 1.89 |
| **Logarithmic Growth** | Logarithmic Upward | 0.70 | 1.91 |
| | Logarithmic Goes Up | 0.73 | 2.13 |
| **Logarithmic Decay** | Logarithmic Goes Down | 0.79 | 1.88 |
| | Logarithmic Downward | 0.83 | 1.75 |
| **Linear Growth and Decay** | Linear Goes Up and Linear Goes Down | 0.71 | 1.93 |
| **Linear Decay and Growth** | Linear Goes Down and Linear Goes Up | 0.73 | 1.51 |
| **Increase Amplitude** | Raise Amplitude | 0.65 | 2.12 |
| **Decrease Amplitude** | Reduce Amplitude | 0.63 | 2.38 |

Table 8: Zero-shot performance of AutoPrompter across various instructed actions.

Table 9 and Table 10 show the average reconstruction losses for different datasets when evaluated with input lengths of 512 and 2048, respectively.

| Dataset | Average Reconstruction Loss | Dataset | Average Reconstruction Loss |
|---------|------------------------------|---------|------------------------------|
| ETTh1 | 0.028 | National Illness | 0.039 |
| Traffic | 0.047 | Exchange Rate | 0.005 |
| Weather | 0.007 | ETTm1 | 0.019 |
| ETTh2 | 0.021 | Electricity | 0.035 |
| ETTm2 | 0.019 | Lorenz Time Series | 0.010 |

Table 9: Reconstruction losses for various datasets: Zero-shot with input length = 512.

| Dataset | Average Reconstruction Loss | Dataset | Average Reconstruction Loss |
|---------|------------------------------|---------|------------------------------|
| ETTh1 | 0.028 | National Illness | 0.039 |
| Traffic | 0.047 | Exchange Rate | 0.005 |
| Weather | 0.007 | ETTm1 | 0.019 |
| ETTh2 | 0.021 | Electricity | 0.035 |
| ETTm2 | 0.019 | Lorenz Time Series | 0.010 |

Table 10: Reconstruction losses for various datasets: Zero-shot with input length = 2048.

## F.2 VISUALIZATION OF RECONSTRUCTION ABILITY

Figures 14 and 18 depict the model's reconstruction ability across various datasets for input lengths of 512 and 2048, respectively.

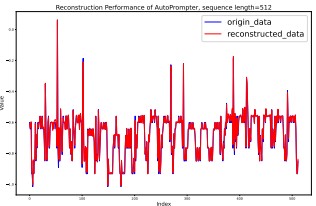
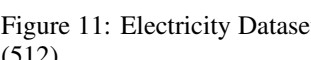
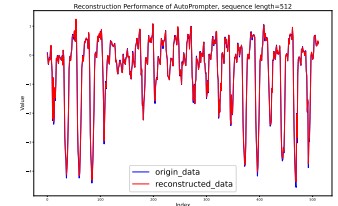
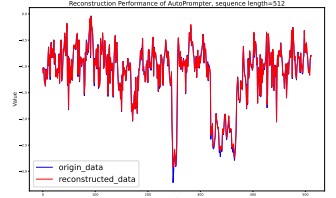

Figure 11: Electricity Dataset (512)    Figure 12: ETTh1 Dataset (512)    Figure 13: ETTh2 Dataset (512)

Figure 14

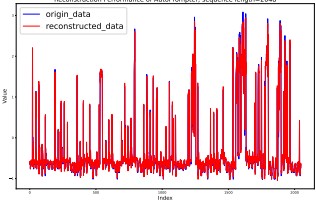
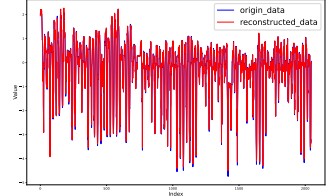
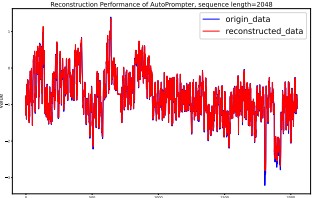

Figure 15: Electricity Dataset (2048)    Figure 16: ETTh1 Dataset (2048)    Figure 17: ETTh2 Dataset (2048)

Figure 18

## F.3 COMPARISON WITH OTHER BASELINES

To further evaluate the performance of AutoPrompter, we compare it against other baseline methods. Figure 19 illustrates how AutoPrompter performs in terms of reconstruction loss when compared to various baselines on the ETTh1 dataset.

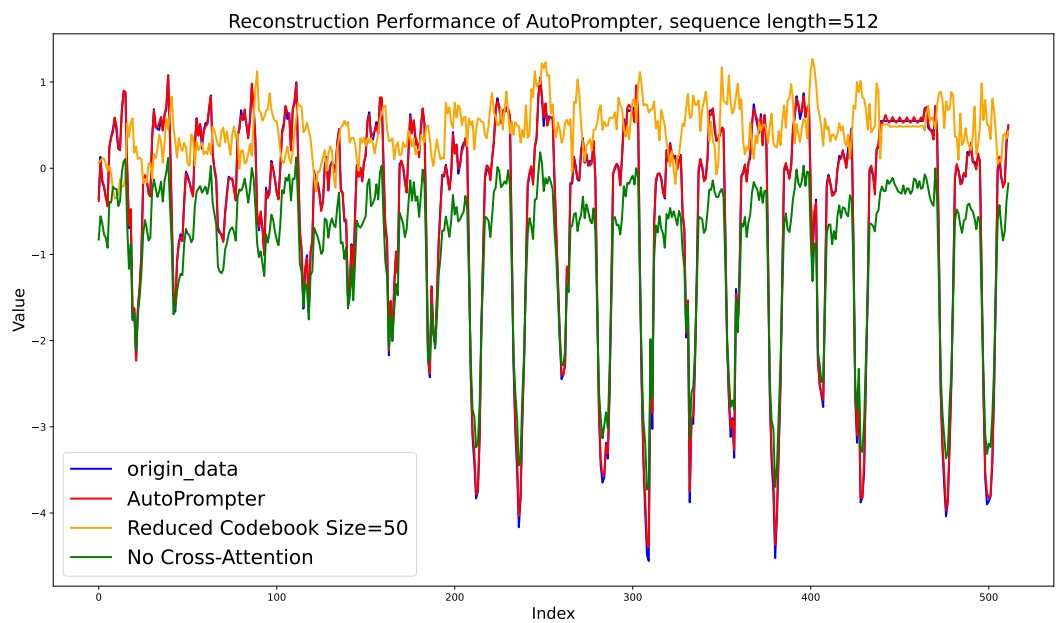

Figure 19: Comparison of AutoPrompter and other baselines on the ETTh1 dataset.

### F.4 Performance with Different Compressed Codebook Sizes

We also examine the effect of different compressed codebook sizes on the total reconstruction loss. As shown in Figure 20, the performance of the AutoPrompter is sensitive to the size of the compressed codebook.

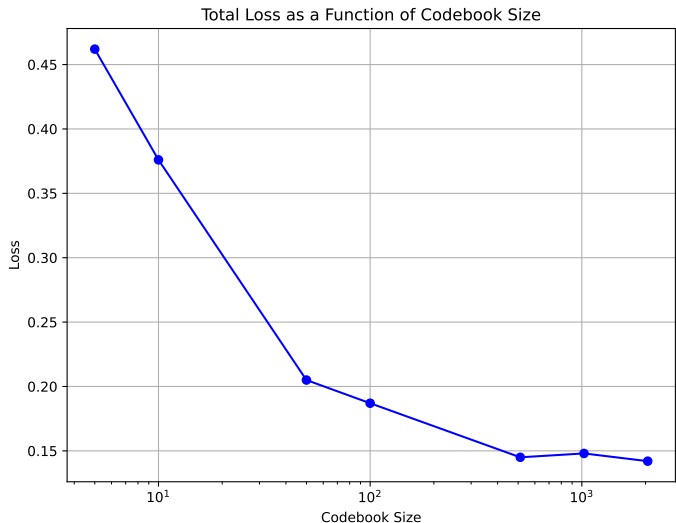

Figure 20: Total loss performance with varying compressed codebook sizes.

## G Dealing with Long Sequences

In this section, we discuss the influence of different lengths of prompts. In practical use, users may input long sequences, but only a few words may be crucial for instructing time-series prediction. In our TITSP model, the attention query is the aligned embedding of the time-series, and the key and value are the embeddings of the prompt.

We study three different settings:

- **Case 1:** The input prompt is only about the instruction, with a length of about 2 to 5 words.
- **Case 2:** The input prompt includes both the instruction and dataset description, with a length of about 30 to 40 words.
- **Case 3:** The input prompt includes the instruction, dataset description, and randomly initialized tokens, with a length greater than 50 words, totaling about 100 words.

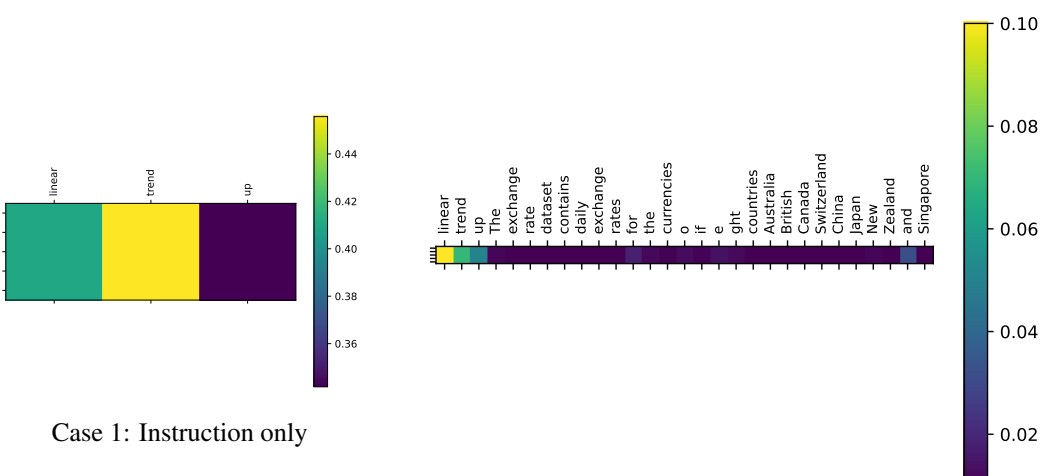

Case 1: Instruction only

Case 2: Instruction and dataset description

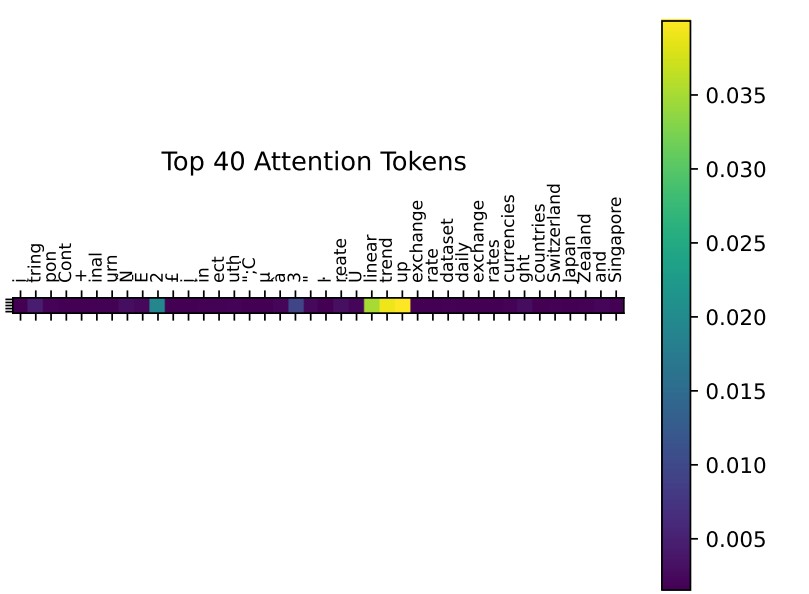

Case 3: Random tokens with instruction and dataset description, showing top 40 tokens' attention weights

Figure 21: Attention matrix study when dealing with long sequences

From the attention maps, we can identify that TITSP can effectively handle long sequences and extract essential words that are crucial for determining the prediction result.

To further explore the performance of TITSP, we compute the compliance rate and MSE performance with different input sequence lengths. In this case, we choose the instruction "Linear trend

up" and calculate the compliance rate and MSE for these three different settings. The following figure visualizes the results.

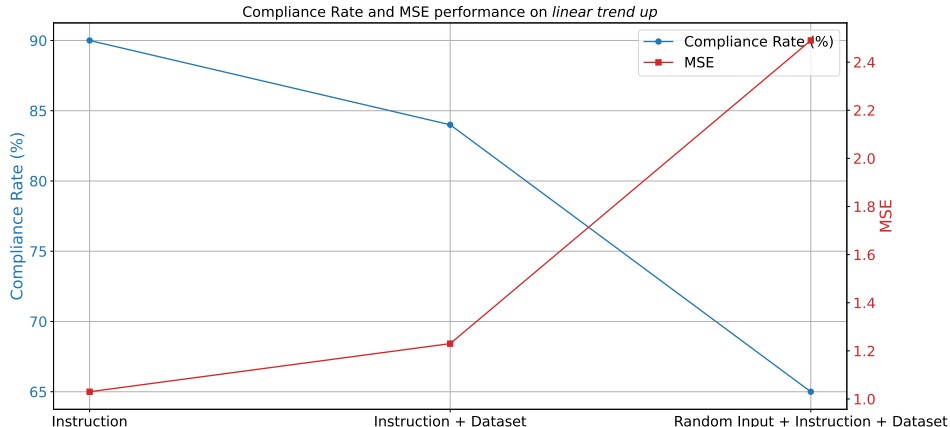

Figure 22: MSE and Compliance rate performance with different input sequence lengths

Our model can effectively extract essential information when dealing with long input sequences, performing well in this zero-shot situation. However, as the figure shows, the performance worsens as the input sequence length increases.

# H    SENSITIVITY OF $\lambda_{\text{RECON}}$

In this section, we discuss the sensitivity of $\lambda_{\text{recon}}$. As described in Section 4.2, there are two parts to the loss function, and $\lambda_{\text{recon}}$ is set to balance them. We choose $\lambda_{\text{recon}}$ to be 0.1, 1, 5, 10, and 20 to evaluate the performance of the data reconstruction loss on the ETTh1 dataset. The following figure visualizes the results, and we finally choose 10 as the value of $\lambda_{\text{recon}}$.

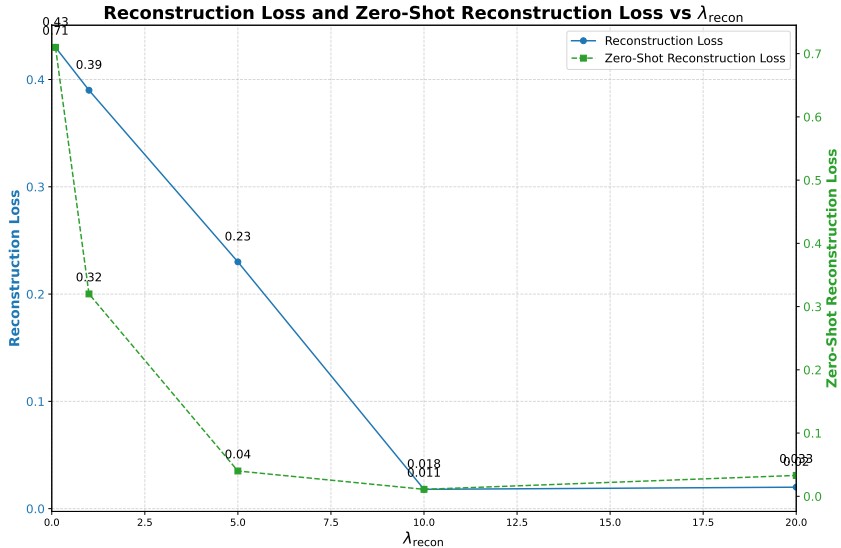

Figure 23: Reconstruction and zero-shot ability with different $\lambda_{\text{recon}}$

# I    DETAILED EXPERIMENTAL SETUP FOR THE PROPOSED METHOD

To ensure the quality of data generation, we utilize different datasets for different instructions. Table 6 outlines which dataset is used to generate new data based on the corresponding instructions.

Additionally, Section C provides details about how we select the hyperparameters and functions. Here, we summarize the process based on the original dataset.

- The sequence length is set to 192, with the first 96 points used as the input $x$.

- For the prediction target $y$, the values are adjusted according to the specific instructions.

Table 11: Datasets with different instructions and the number of training examples.

| Dataset | Instruction | Count |
|---|---|---|
| ETTH1 | increase amplitude
decrease amplitude
keep stable | 14307 |
| ETTH2 | increase amplitude
decrease amplitude
keep stable | 14307 |
| Weather | linear trend up
linear trend down
exponential growth
exponential decay
logarithmic growth
logarithmic decay
keep stable
linear growth and linear decay
linear decay and linear growth | 52603 |
| Exchange rate | linear trend up
linear trend down
exponential growth
exponential decay
logarithmic growth
logarithmic decay
keep stable
linear growth and linear decay
linear decay and linear growth | 7207 |
| Electricity | linear trend up
linear trend down
exponential growth
exponential decay
logarithmic growth
logarithmic decay
keep stable
linear growth and linear decay
linear decay and linear growth | 26210 |
| ETTm1 | increase amplitude
decrease amplitude
keep stable | 57507 |
| ETTm2 | increase amplitude
decrease amplitude
keep stable | 57507 |
| Traffic | increase amplitude
decrease amplitude
keep stable | 16476 |

The detailed function of each instruction is described in Section C. We select the instruction for each dataset based on the original features of the dataset.

## DETAILS ABOUT THE CONVOLUTION LAYERS AND PATCHING

### PATCHING PROCESS

The input time series are divided into smaller sequences of length 96, similar to the approach used in PatchTST (Nie et al., 2022).

### ENCODER ARCHITECTURE

The encoder in **AutoPrompter** consists of:

- Three 1-dimensional convolutional layers.
- One residual connection applied to the first layer.
- **ReLU** activation function after the second and third convolutional layers.
- A linear layer projects features to a dimensionality of 1536, matching the language model embedding size.

The encoder structure is summarized in Table 12. Layers with a stride of 2 compress the input sequence by a factor of 4, meaning every four points are combined into one embedding in the compressed semantic space.

Table 12: Encoder Configuration

| Layer | Input Channels | Output Channels | Kernel Size | Stride | Padding |
|-------|----------------|-----------------|-------------|--------|---------|
| Convolution Layer 1 | 1 | 128 | 4 | 2 | 1 |
| Convolution Layer 2 | 128 | 256 | 4 | 2 | 1 |
| Convolution Layer 3 | 256 | 256 | 3 | 1 | 1 |

### DECODER ARCHITECTURE

The decoder consists of:

- Three convolutional layers.
- One residual connection.

The decoder structure is shown in Table 13.

Table 13: Decoder Configuration

| Layer | Input Channels | Output Channels | Kernel Size | Stride | Padding |
|-------|----------------|-----------------|-------------|--------|---------|
| Convolution Layer 1 | 1536 | 256 | 3 | 1 | 1 |
| Convolution Layer 2 | 256 | 128 | 4 | 2 | 1 |
| Convolution Layer 3 | 128 | 1 | 4 | 2 | 1 |

### DECODER CONNECTED TO LLM OUTPUT

The decoder connecting to the LLM output consists of a single 1-dimensional convolutional layer (Table 14). Since the original time series is compressed by a factor of 4, there are 24 embeddings of size 1536. The input channel is the product of 24 and 1536 after flattening.

Table 14: Decoder-LLM Configuration

| Layer | Input Channels | Output Channels | Kernel Size | Stride | Padding |
|-------|----------------|-----------------|-------------|--------|---------|
| Convolution Layer 1 | 36864 | 96 | 3 | 1 | 1 |

## HYPERPARAMETERS USED

The hyperparameters used in the experiments are summarized in Table 15. Early stopping is applied if no improvement is observed on the test set for more than 10 epochs. For the electricity and traffic datasets, the number of epochs is reduced to 5 due to faster convergence.

Table 15: Hyperparameter Configuration

| Hyperparameter | Value |
|---|---|
| Learning Rate | 5.00E-04 |
| Batch Size | 16 |
| Epochs | 20 (default), 5 (electricity/traffic) |
| Optimizer | Adam |

## IMPLEMENTATION DETAILS FOR QWEN4MTS, UNITIME AND LLAMA-8B-INSTRUCT

### QWEN4MTS (FROM GPT4MTS)

Following GPT4MTS (Jia et al., 2024):

- Word Embedding, Position Embedding, Add & Norm, and Output Linear Layer are trainable.
- The embedding size is set to 1536, consistent with Qwen.
- The learning rate is set to `5e-5`.

### UNITIME

Following UniTime (Liu et al., 2024b):

- A binary indicator is used to generate the mask.
- Time-series embeddings are concatenated with sentence embeddings.
- The hidden dimension is set to 1536 (matching Qwen), with `n_embd` set to 1536.
- The mask rate is 0.5, and the learning rate is set to `1e-4`.
- The lightweight transformer is replaced with Qwen LLM.
- All experiments are trained for 10 epochs.

### LLAMA-3.1-8B-INSTRUCT

Prompt Design for Llama-3.1-8B-instruct (Dubey et al., 2024):

The prompt include task description, instruction and specific input number. The following is an example.

- "[Task Description]: forecast the next 96 steps given the previous 96 steps information."
- "[Instruction]:the prediction should follow 'Linear Growth'"
- "[Input Number]: 0.173, 0.125, ..."

## J   WHAT REALLY HAPPENED ON OCTOBER 10, 2008?

The sharp drop in the GBP/USD exchange rate on October 11, 2008, can be traced to events on October 10, 2008, when global financial markets experienced a massive sell-off, exacerbating the effects of the ongoing financial crisis.

On October 10, 2008, the Dow Jones Industrial Average plunged 679 points (or about 7.3%) by the end of the trading day, after dipping as much as 800 points during intra-day trading. European markets saw even steeper declines, with the FTSE 100 (UK's benchmark index) dropping around 8.9% and Germany's DAX falling 7%. In Asia, Japan's Nikkei 225 dropped 9.6%, adding to the already severe financial instability.

Despite the UK government's £500 billion bailout package announced on October 8 to rescue major banks like Royal Bank of Scotland and Lloyds TSB, concerns over the health of the global financial system persisted. Investors began fleeing to safer assets, driving demand for the U.S. dollar as a safe haven, which in turn weakened the British pound.

In addition to the stock market crashes, the growing fear of a deep global recession weighed heavily on the British pound, especially because the UK economy was seen as particularly vulnerable due to its dependence on the financial sector. This combination of stock market turmoil, concerns over the UK's banking system, and the safe-haven demand for the U.S. dollar caused the sharp decline in the GBP/USD exchange rate on October 11, 2008 (Times, 2008).

