# OpenReview forum: "INSTRUCTION-FOLLOWING LLMS FOR TIME SERIES PREDICTION: A TWO-STAGE MULTIMODAL APPROACH"
_ICLR.cc/2025/Conference — Submitted to ICLR 2025_

### Official Review · Reviewer_We4d · 2024-10-29

**Soundness:** 2
**Presentation:** 3
**Contribution:** 2
**Rating:** 5
**Confidence:** 4

**Summary:**

The paper proposes a novel two-stage for multimodal forecasting through historical data and textual cues that are useful for LLM-based forecasters. The multimodal framework is evaluated on numerous multimodal forecasting tasks. The paper provides a setup to include expert opinions for a forecasting problem.

**Strengths:**

The strengths include the relevance of the problem of text-aided forecasting and the novelty of the prompting method. The methodology section is comprehensive and well-described, and the techniques and experiments have been explained in detail and are easy to follow. The figures convey the overall idea and highlight the improvements over the no-instruction setup.

**Weaknesses:**

The primary weaknesses of the paper are as follows:

1. **Incomplete Literature Coverage**: Section 2.2 does not fully address relevant multimodal forecasting models, omitting key references such as UniTime ([https://dl.acm.org/doi/10.1145/3589334.3645434](https://dl.acm.org/doi/10.1145/3589334.3645434)).

2. **Limited Comparative Analysis**: The results lack sufficient comparison with other multimodal forecasting models, reducing insight into how the proposed method performs relative to similar approaches.

3. **Insufficient Dataset Description**: Essential dataset details, including sample counts, history length, and forecasting horizon, are not provided. Additionally, the impact of the forecasting horizon on prediction quality remains underexplored.

4. **Simplistic Experimental Instructions**: The experimental instructions are overly simplistic, failing to reflect realistic scenarios. The limited set of training instructions may also suggest that simpler alternatives for instruction embedding could have been more effective.

5. **Circular  Evaluation**: The evaluation datasets have been tailored from existing datasets based on the training instructions intended for evaluation, which creates a circular reasoning issue that undermines the reliability of the evaluation setup.  A similar statement about the order compliance rate metric can also be made.

**Minor Issues:**

1. The paper inconsistently uses closing quotes (") instead of opening quotes (``) in multiple locations, including but not limited to lines 197, 203, and 213.

2. Textual citations, rather than parenthetical citations, would be more suitable for the references in lines 117 to 128, enhancing the readability and flow of the text.

3. Appropriate citations are not provided for the original dataset sources.

**Questions:**

Questions:
1. The choice of order compliance rate as an evaluation metric is intriguing. This metric appears specifically tailored to the instructions outlined in the paper, which may limit its applicability to real-world scenarios. Could you clarify the advantages this metric offers over existing metrics for evaluating forecasting performance?

Suggestions:

- Benchmark results against a broader selection of existing multimodal forecasting models to enhance comparative insights.
- Include a detailed discussion of the dataset, covering aspects such as sample size, history length, and forecasting horizon.
- If feasible, incorporate more complex textual cues in the experiments to better reflect real-world forecasting challenges.

---

> ### Author Response · Authors · 2024-11-18
> **Response to Reviewer We4d**
>
> ### Comment 1:
> *Major issues*
>
> **Response:**
> We thank the reviewer for their insightful comments. We address each of the points raised as follows:
>
> - **Incomplete Literature Coverage:**
>   We appreciate the reviewer pointing out the omission of key references such as UniTime. We have incorporated this important work, along with other relevant multimodal forecasting models, into the updated version of the paper. The related work section has been expanded to provide a more comprehensive overview of the field and to better position our contribution in relation to existing approaches.
>
> - **Limited Comparative Analysis:**
>   We appreciate the reviewer’s insightful feedback regarding the need for broader comparisons with other multimodal forecasting models. To address this concern, we have expanded our comparisons to include additional multimodal models, particularly in scenarios where descriptive text is provided alongside time series data. Notably, we have included comparisons with GPT-4TS, TimeLLM, and purely time-series-based models (_Table 5_), and our method outperforms these models in the tasks considered, demonstrating its high performance even outside the scope of instruction-based tasks. Additionally, we present a detailed evaluation of our method against UniTime and Qwen4MTS in _Table 2 (Page 9)_ to further address the reviewer’s concerns.
>
> - **Insufficient Dataset Description:**
>   We apologize for the lack of detail regarding the datasets. We have taken care to include all relevant dataset details—such as sample counts, history length, and forecasting horizon—in the updated manuscript. A detailed analysis is presented in Section I of the Appendix.
>
> - **Simplistic Experimental Instructions:**
>   While the instructions considered in our work may seem simple, they represent an essential first step toward more complex scenarios where complete document instructions can be provided. Our research serves as a foundational effort, demonstrating that even with simple cases, several challenges must be addressed using existing algorithms like TimeLLM. We show that these challenges can be effectively managed with a specifically designed architecture.
>
>   Although there is room for improvement in handling instructions, our paper already considers scenarios where the test text instructions differ from those used during training. These instructions are somewhat complex, combining several base instructions, and our methods demonstrate good generalization capabilities (_Table 3_). This success suggests promising perspectives toward the ultimate goal of tackling any instruction, regardless of complexity.
>
> - **Circular Evaluation:**
>   We appreciate the reviewer’s feedback regarding the evaluation datasets and the potential for circular reasoning. To address this concern, we highlight that in _Table 3_, we provide an assessment of the generalization capabilities of our model. In this evaluation, the training and test instructions differ significantly, which we believe is a fair and robust way to evaluate the model's ability to handle instructions adequately.
>
>   This approach ensures that our model is tested on scenarios not explicitly covered during training, providing a more reliable measure of its performance in real-world applications. We are confident that this evaluation demonstrates the model's generalization capabilities and addresses the reviewer’s concerns about the reliability of our evaluation setup.
>
> ---
>
> ### Comment 2:
> *Minor issues*
>
> **Response:**
> Thank you for your detailed feedback on our paper. We appreciate your time and effort in providing these valuable comments. We will take each of your suggestions into account in the updated version of the paper.
>
> - Regarding the inconsistent use of closing quotes (`"`) instead of opening quotes (`“`) in multiple locations, including but not limited to lines 197, 203, and 213, we will ensure that the correct quotation marks are used throughout the manuscript.
> - We agree with your suggestion to use textual citations rather than parenthetical citations for the references in lines 117 to 128. This will enhance the readability and flow of the text.
> - Additionally, we will provide appropriate citations for the original dataset sources.
>
> Thank you once again for your constructive feedback. We look forward to addressing these points in the revised version of the paper.
>
> ---

---

> > ### Author Response · Authors · 2024-11-18
> > **Response to Reviewer We4d**
> >
> > ### Comment 3:
> > *The choice of order compliance rate as an evaluation metric is intriguing. This metric appears specifically tailored to the instructions outlined in the paper, which may limit its applicability to real-world scenarios. Could you clarify the advantages this metric offers over existing metrics for evaluating forecasting performance?*
> >
> > **Response:**
> > We appreciate the reviewer’s thoughtful question. The order compliance rate was specifically chosen as an evaluation metric because the primary goal of our work is to assess how well the model adheres to the given textual instructions. Since we are the first to propose text instruction-based forecasting, existing metrics for traditional forecasting tasks may not fully capture the performance of a model that must follow complex, hypothetical instructions.
> >
> > The compliance rate, therefore, offers a tailored and effective way to measure this adherence. While it may seem specific to our setting, we believe it is a novel and valuable metric for this new approach. Its design enables us to quantify how well the model aligns with textual instructions, which is central to the novelty of our framework. We hope this clarifies why this metric is appropriate and meaningful in the context of our work.
> >
> > ### Suggestions of the reviewer:
> > *Benchmark results against a broader selection of existing multimodal forecasting models to enhance comparative insights.
> > Include a detailed discussion of the dataset, covering aspects such as sample size, history length, and forecasting horizon.
> > If feasible, incorporate more complex textual cues in the experiments to better reflect real-world forecasting challenges.*
> >
> > **Response:**
> > We thank the reviewer for their valuable suggestions. In the updated version of the paper (_Table 2, page 9_), we have included an exhaustive comparison with other methods tailored for text-instruction-based forecasting, including Qwen4MTS and UniTime, as also requested by Reviewer mT1k. Additionally, we extend the evaluation to descriptive tasks, where text serves as a description rather than an instruction. In the Appendix (_Table 5_), we compare our method against several benchmarks (including GPT4TS, TimeLLM, and time series-based forecasters) and show that, even without textual instructions (with descriptive texts about the task), our approach outperforms other models, demonstrating its broader applicability.
> >
> > We have also added more details on the datasets used, including sample size, history length, and forecasting horizon in Section I of the Appendix.
> >
> > Further More, our method could handle long sequence input which is shown in appendix G, and it shows a good attention map that it can extract key words regarding to the user instruction so that it could be used in real life.

---

> ### Author Response · Authors · 2024-11-21
> **Response to Reviewer We4d**
>
> Dear reviewer:
>     I wonder if our response can solve your concerns! Thank you!

---

> > ### Comment · Reviewer_We4d · 2024-11-23
> >
> > Thank you for addressing my comments and revising the paper.
> >
> > While most of my concerns have been addressed, I still have some questions regarding the core contribution of this work. In Sections 2.2, the authors claim to present a _novel framework_ for forecasting that leverages textual instructions and demonstrate its superior performance over existing frameworks in Table 2. However, the claimed novelty of this framework compared to existing methodologies remains unclear. I request the authors to further **elaborate on the framework's uniqueness as compared to the existing methods** and include the **parameter counts for both their model and the benchmarks** to confirm that the improvements are not merely due to higher computational resources.
> >
> > Furthermore, despite the authors' appreciating the raised typographical issues, such issues have continued into the revised sections, with some of them listed below:
> > 1. Incorrect quotations- line 113
> > 2. Incorrect parenthetical citations- line 127
> > 3. Spelling errors- line 107 (_success_ - _sucess_)
> >
> > Such typos, though minor, are numerous enough to raise concerns about the overall credibility of the paper. For now, I will maintain my current score.

---

### Official Review · Reviewer_YdJR · 2024-10-30

**Soundness:** 2
**Presentation:** 3
**Contribution:** 3
**Rating:** 5
**Confidence:** 3

**Summary:**

The paper introduces Text-Informed Time Series Prediction (TITSP), a novel two-stage framework that enhances time series forecasting by integrating domain-specific textual information. The paper demonstrates that TITSP significantly outperforms traditional and existing multimodal approaches, improving both predictive accuracy and interpretability.

**Strengths:**

1. The paper presents a novel approach to time series forecasting by integrating textual instructions, which is a creative extension of existing multimodal time series models. The introduction of a two-stage framework and the focus on instruction-based forecasting address a significant gap in the field.
2. The paper is well-written and logically organized. The figures and tables are clear and effectively support the text. The problem formulation and the description of the methodology are easy to follow.

**Weaknesses:**

1. Given the synthetic data generation process, how can the authors ensure that there is no data leakage between the text data and forecasting targets? Could the authors provide a detailed explanation of the data generation process to address this concern?
2. How practical is the proposed approach in real-world scenarios where textual instructions may not always be available or may be ambiguous? Could the authors discuss the potential limitations and challenges in deploying TITSP in practical applications?
3. Has the model been tested on any other multimodal time series analysis tasks beyond forecasting? If not, what are the potential challenges in applying TITSP to other tasks?

**Questions:**

Please see the weaknesses.

**Details Of Ethics Concerns:**

The paper does not raise any significant ethical concerns.

---

> ### Author Response · Authors · 2024-11-18
> **Response to Reviewer YdJR**
>
> ### Comment 1:
> *Given the synthetic data generation process, how can the authors ensure that there is no data leakage between the text data and forecasting targets? Could the authors provide a detailed explanation of the data generation process to address this concern.*
>
> **Response:**
> We thank the reviewer for raising this important question. While the concern about data leakage is valid in many contexts, it is not a central issue in our case. The primary goal of our work is to assess the model's adherence to specific textual instructions rather than predict the target based solely on the time series data. To clarify, consider three samples with identical context length: a deterministic machine learning model would typically produce the same forecast for these samples. However, by adding textual instructions that specify a particular scenario or condition, we introduce a new layer of information that the model must adhere to. This is not a data leakage problem but rather a way of interacting with the model through different hypothetical scenarios.
>
> The compliance rate is explicitly defined to measure how well the model follows these instructions while preserving the underlying time series structure. Thus, the model’s ability to follow instructions is the focus, rather than predicting targets based solely on historical data. This being said, while the focus of the paper is on text-instructed problems, we also perform experiments in the Appendix on other types of data where the text describes the task (e.g., domain, forecasting type, features). In these cases, the dataset contains no leakage, and our proposed algorithm outperforms the state-of-the-art.
>
> ---
>
> ### Comment 2:
> *How practical is the proposed approach in real-world scenarios where textual instructions may not always be available or may be ambiguous? Could the authors discuss the potential limitations and challenges in deploying TITSP in practical applications?*
>
> **Response:**
> We appreciate the reviewer’s thoughtful question. As discussed in our response to the first comment, the primary purpose of our approach is to evaluate how well the model adheres to specific textual instructions in controlled scenarios. While textual instructions are central to this evaluation, we acknowledge that in real-world applications, such instructions may not always be available or could be ambiguous.
>
> One potential limitation is the reliance on clear, actionable instructions, which may not always be feasible in dynamic or unstructured environments. Additionally, the model’s performance may be affected by the quality and specificity of the textual input. However, our framework is designed to handle a wide range of instruction formats and adapt to different hypothetical scenarios, making it flexible for practical deployment. We also envision that the model could be augmented with supplementary mechanisms (e.g., user feedback loops or clarification prompts) to address ambiguity in real-world use cases.
>
> ---
>
> ### Comment 3:
> *Has the model been tested on any other multimodal time series analysis tasks beyond forecasting? If not, what are the potential challenges in applying TITSP to other tasks?*
>
> **Response:**
> We appreciate the reviewer’s question. While our model has primarily been tested for forecasting tasks, extending it to other multimodal time series analysis tasks such as classification presents some challenges. In classification, the output is often constrained to predefined labels, which limits the flexibility needed to explore different hypothetical scenarios through textual instructions. This makes it difficult to leverage the full potential of our approach in such tasks.
>
> However, as demonstrated in the Appendix, our framework can be extended to scenarios where instead of instructions, we incorporate other types of additional information related to the task at hand. In these cases, our model still outperforms existing methods, suggesting that the framework has potential beyond forecasting, even for tasks with more constrained output spaces. Furthermore, we believe that imputation tasks could be a natural extension of our framework, as it can easily accommodate missing data by conditioning on other available information, showing that our approach is adaptable to different problem settings.

---

> > ### Author Response · Authors · 2024-11-18
> > **Response to Reviewer YdJR - result table**
> >
> > For convenience, we also provide table 5 in the paper, which is about the test results. We add more experiments to show the effectiveness of our algorithm.
> >
> > ### Table: Comparison of Compliance Rate (CR) and MSE for TITSP, Time-LLM, Qwen4MTS, UniTime, and Llama-3.1-8B across various instructed actions with highlighted best (in **bold**) and second-best (in _underlined_) results.
> >
> > | **Instruction**                 | **TITSP (CR)** | **TITSP (MSE)** | **Time-LLM (CR)** | **Time-LLM (MSE)** | **Qwen4MTS (CR)** | **Qwen4MTS (MSE)** | **UniTime (Qwen) (CR)** | **UniTime (Qwen) (MSE)** | **Llama-3.1-8B (CR)** | **Llama-3.1-8B (MSE)** |
> > |---------------------------------|----------------|-----------------|-------------------|--------------------|-------------------|--------------------|-------------------------|-------------------------|------------------------|------------------------|
> > | Linear Growth and Linear Decay  | **0.83**       | **1.15**        | 0.38             | 3.45               | _0.69_            | _1.90_             | 0.54                   | 2.73                   | 0.32                  | 4.95                  |
> > | Linear Growth and Linear Decay  | **0.79**       | **1.17**        | 0.49             | 2.85               | **0.79**          | _1.34_             | 0.57                   | 2.28                   | 0.41                  | 2.80                  |
> > | Linear Trend Up                 | _0.90_         | **1.03**        | 0.63             | 1.71               | 0.76              | _1.08_             | 0.63                   | 1.65                   | **0.91**              | 1.15                  |
> > | Linear Trend Down               | **0.87**       | **0.88**        | 0.64             | 1.55               | 0.71              | 1.36               | 0.51                   | 1.59                   | _0.85_                | _0.92_                |
> > | Exponential Growth              | **0.89**       | **1.33**        | 0.58             | 2.59               | _0.63_            | _2.07_             | 0.60                   | 2.38                   | 0.58                  | 2.35                  |
> > | Exponential Decay               | **0.84**       | **1.25**        | 0.56             | 2.26               | 0.67              | 2.10               | _0.69_                 | _2.05_                 | 0.46                  | 2.39                  |
> > | Keep Stable                     | **0.98**       | _0.35_          | 0.76             | 0.76               | 0.93              | 0.48               | 0.83                   | 0.62                   | _0.95_                | **0.33**              |
> > | Decrease Amplitude              | **0.90**       | _0.91_          | 0.85             | 1.04               | **0.90**          | **0.84**           | 0.79                   | 1.09                   | 0.52                  | 1.89                  |
> > | Increase Amplitude              | **0.94**       | **0.94**        | 0.79             | 1.20               | _0.89_            | _0.96_             | 0.81                   | 1.03                   | 0.75                  | 1.35                  |
> > | Logarithmic Growth              | _0.77_         | _1.65_          | 0.49             | 2.31               | **0.79**          | **1.55**           | 0.60                   | 1.73                   | 0.55                  | 1.94                  |
> > | Logarithmic Decay               | **0.83**       | **1.68**        | 0.48             | 2.19               | _0.81_            | _1.69_             | 0.67                   | 2.04                   | 0.63                  | 2.60                  |

---

> ### Author Response · Authors · 2024-11-21
> **Response to Reviewer YdJR**
>
> Dear reviewer: I wonder if our response can solve your concerns! Thank you!

---

> > ### Comment · Reviewer_YdJR · 2024-11-26
> >
> > I appreciate the authors' response. However, the issue of data leakage and the resulting concerns regarding practical applicability remain unresolved.
> >
> > I understand the authors' claim that their model can effectively capture textual instructions about future time series, outperforming previous models. Nonetheless, in real-world scenarios, it is highly improbable that we would have access to highly accurate future textual data. This implies that the textual information representing future trends in practical applications is likely to be significantly inaccurate, resulting in a substantial difference between the training and testing datasets of the framework and real-world conditions. Even if we hypothetically assume that we could reliably obtain highly accurate textual instructions about the future, would we then only require manual intervention based on these precise descriptions to make predictions?
> >
> > In summary, I am concerned that there is a substantial disconnect between the future information used for training and testing and the future textual descriptions that will be available in practical applications, which raises questions about the actual efficacy of the proposed framework.

---

> > > ### Author Response · Authors · 2024-11-26
> > >
> > > Thank you for your insightful response. We truly appreciate your feedback, which encourages us to further elaborate on our approach.
> > >
> > > TITSP is designed to revolutionize time-series prediction by making it interactive. In contrast to traditional deep learning methods, which often fail in real-world applications due to their reliance solely on historical data, TITSP empowers users to actively participate in the prediction process. Deep learning models tend to learn only from past patterns, requiring users to repetitively re-engineer features and retrain models to have better performance.
> > >
> > > In our framework, we enable users to engage directly with the prediction process, not by assuming perfect textual descriptions, but by allowing them to inject their expert judgment and professional knowledge. This integration of human insight into time-series forecasting marks a significant departure from conventional methods, creating a more dynamic and adaptive approach to prediction.

---

### Official Review · Reviewer_GGqR · 2024-10-30

**Soundness:** 2
**Presentation:** 1
**Contribution:** 2
**Rating:** 3
**Confidence:** 3

**Summary:**

The paper presents Text-Informed Time Series Prediction (TITSP), a multimodal framework that integrates textual context with time series data using Large Language Models (LLMs). The approach involves two stages: AutoPrompter, which aligns time series data with text embeddings, and a refinement stage that incorporates task-specific textual instructions to enhance prediction accuracy and interpretability. While TITSP proves particularly effective for context-rich forecasting tasks, by demonstrating improved performance under specific settings against some other methods.

**Strengths:**

- A novel two-stage framework for integrating temporal and textual data.
- A data generation workflow for instruction-based forecasting, compatible with LLMs.
- Comprehensive ablation studies and comparative evaluations demonstrating the effectiveness of TITSP.

**Weaknesses:**

- **Technical Contributions are Incremental**  The proposed approach lacks significant technical innovation. Integrating LLMs with time series is an incremental step rather than a groundbreaking contribution. The use of cross-attention and VQ-VAE offers no substantial improvement beyond established techniques.
- **Poor Structure and Clarity**  The paper is poorly organized, with unclear explanations and an incoherent flow. The motivation and rationale for the proposed method are inadequately communicated, and critical components like AutoPrompter are explained in a convoluted manner, hindering comprehension.
- **Inadequate Experiments**  Experimental validation is weak, relying heavily on synthetic datasets that limit the assessment of practical applicability. Comparisons to related state-of-the-art methods are lacking, and statistical significance testing is absent, making it difficult to validate the performance claims.
- **Superficial Related Work**  The related work section lacks depth and fails to properly differentiate the contribution from prior research. Key works are missing or insufficiently discussed, weakening the justification for originality.
- **Numerous Typos and Lack of Polish**  Frequent typos (e.g. citation mistaches in line 54-55), poorly formatted figures(fig. 6), and poorly constructed tables suggest a lack of careful proofreading, which detracts from the overall quality and credibility of the paper.
- **Insufficient Practical Insights**  The claimed interpretability through textual integration lacks demonstration. There are no real-world examples showing how domain experts would benefit from these insights, making the practical value of TITSP unclear.

**Questions:**

- **How would the proposed model perform without access to textual inputs or under noisy conditions?** If textual instructions are incomplete, inconsistent, or contain noise, how would the model's performance be affected? This scenario is particularly relevant in high-stakes areas like finance, where decision-making often involves dealing with imperfect information. What measures have been taken to ensure robustness against these issues, which are common in real-world data?
- **How does the proposed framework address interpretability in practice?** The paper claims that incorporating textual instructions enhances interpretability, but there are no concrete demonstrations of how this contributes to meaningful insights for domain experts. Could you provide explicit examples or user studies that validate this claim? Without such evidence, how can the claim of improved interpretability be substantiated?

---

> ### Author Response · Authors · 2024-11-18
> **Response to Reviewer GGqR**
>
> Thank you for your precious comments!
>
> ### Comment 1:
> *Technical Contributions are Incremental*
> _The proposed approach lacks significant technical innovation. Integrating LLMs with time series is an incremental step rather than a groundbreaking contribution. The use of cross-attention and VQ-VAE offers no substantial improvement beyond established techniques._
>
> **Response:**
> We appreciate the reviewer’s feedback. While it is true that certain components of our architecture, such as cross-attention and VQ-VAE, are established techniques, our contribution lies in the development of a novel methodological framework. This framework includes a tailored data pipeline, innovative architecture design, and a comprehensive evaluation approach, all specifically geared towards integrating text-based instructions with time series forecasting.
>
> We believe this is a significant contribution because it provides a structured approach to applying language models in the context of time series, which is a growing area of interest with wide applicability in fields like supply and demand forecasting. The ability to define and manipulate hypothetical scenarios through textual instructions opens new avenues for adaptable and context-sensitive forecasting models. We are confident that this framework will be valuable to the community, as it sets a foundation for future work in this space.
>
> ---
>
> ### Comment 2:
> *Poor Structure and Clarity*
> _The paper is poorly organized, with unclear explanations and an incoherent flow. The motivation and rationale for the proposed method are inadequately communicated, and critical components like AutoPrompter are explained in a convoluted manner, hindering comprehension._
>
> **Response:**
> We are sorry to hear that some aspects of the paper were unclear. We greatly value the reviewer’s feedback and would be very interested in a constructive discussion to better understand which specific parts of the motivation and rationale were difficult to follow. In the related work, we have added a paragraph to highlight the main motivation and differences of our work compared to state-of-the-art methods, especially those targeting instruction-based forecasting.
>
> Regarding the reviewer’s concern with the explanation of AutoPrompter, we would like to clarify its purpose: AutoPrompter serves as a bridge that translates the time series data into the text embedding space. By quantizing the time series space, we map it into a compressed semantic space, which may have contributed to some of the complexity in the explanation. We have added additional clarifications in the updated version of the paper (_Page 5_) to ensure this concept is more accessible and the overall flow is clearer.
>
> We appreciate the reviewer’s insights and hope the revised manuscript will address these concerns effectively.
>
> ---
>
> ### Comment 3:
> *Inadequate Experiments, Superficial Related Work, Numerous Typos and Lack of Polish, Insufficient Practical Insights*
>
> **Response:**
>
> **Inadequate Experiments:**
> We acknowledge the reviewer’s concern about the reliance on synthetic datasets. While we agree that real-world data is crucial for evaluating practical applicability, synthetic datasets were used primarily to demonstrate the model’s capacity to handle controlled scenarios where the impact of specific factors can be isolated. We have now included benchmarks against state-of-the-art approaches (Llama-3.1-8B-instruct, Qwen4MTS and Unitime in Table 2). The results are highly stable and reproducible, with substantial performance margins over competitors, which reduce the necessity for statistical significance testing.
>
> **Superficial Related Work:**
> We have expanded the related work section to better differentiate our approach from prior research, particularly in the integration of text and time series. References such as Unitime have been added to strengthen the justification for our originality.
>
> **Insufficient Practical Insights:**
> The interpretability of our framework lies in facilitating interaction between expert users and the model through hypothetical scenarios. For example, the model generates forecasting scenarios based on textual instructions about supply and demand conditions, enabling experts to evaluate potential outcomes. This is particularly useful in fields like supply chain management, where generating and testing "what-if" scenarios through textual inputs offers clear practical benefits.
>
> ---

---

> ### Author Response · Authors · 2024-11-18
> **Response to Reviewer GGqR - question part**
>
> ### Question 1:
> *How would the proposed model perform without access to textual inputs or under noisy conditions? If textual instructions are incomplete, inconsistent, or contain noise, how would the model's performance be affected? This scenario is particularly relevant in high-stakes areas like finance, where decision-making often involves dealing with imperfect information. What measures have been taken to ensure robustness against these issues, which are common in real-world data?*
>
> **Response:**
> We appreciate the reviewer’s question on robustness under imperfect or noisy textual inputs. In “what-if” scenarios, even if expert instructions are incomplete or contain intentional inaccuracies, the model’s primary goal is to follow these inputs accurately. This is often precisely what an expert wants—simply to test various hypothetical scenarios by observing how the model behaves under different instructions, rather than to ensure perfectly accurate instructions. Our model’s high compliance rate shows that it reliably adheres to these inputs, enabling experts to evaluate potential outcomes and behaviors without needing to formalize each scenario mathematically within the framework.
>
> Although handling incomplete or inconsistent text is not the main focus of our work, we recognize its relevance. In the Appendix (_Table 5, Page 18_), we include experiments comparing our model's performance with and without textual inputs. These results show that even in cases without explicit instructional text, our method outperforms purely time-series models, highlighting the value of informative text for forecasting accuracy. This additional evaluation demonstrates the model’s ability to handle various text input scenarios, further affirming its robustness and versatility.
>
> ---
>
> ### Question 2:
> *How does the proposed framework address interpretability in practice? The paper claims that incorporating textual instructions enhances interpretability, but there are no concrete demonstrations of how this contributes to meaningful insights for domain experts. Could you provide explicit examples or user studies that validate this claim? Without such evidence, how can the claim of improved interpretability be substantiated?*
>
> **Response:**
> We thank the reviewer for raising this question on interpretability. The interpretability of our framework primarily comes from its capacity to directly link predictions to textual instructions. For example, if a linear growth is predicted, it can be traced back to specific input instructions, providing clear insight into why a particular behavior was forecasted. Additionally, attention map visualizations (see _Figure 21, Page 27_) reveal that the model highlights relevant keywords from the instructions, further demonstrating its focus on critical components of the input. This not only makes the reasoning process transparent but also allows experts to verify that the model is attending to meaningful terms.
>
> Our framework’s generalization ability also contributes to interpretability, as it shows the model’s capacity to apply learned associations to new contexts, indicating it understands the core instruction beyond specific examples. While we acknowledge the value of explicit user studies, these elements collectively provide substantial interpretability by aligning the model's outputs directly with expert input and highlighting the key instructions that guide predictions.

---

> ### Author Response · Authors · 2024-11-18
> **Response To Reviewer GGqR- result table**
>
> For convenience, we also provide table 5 in the paper about the test results.
>
> ### Table: Comparison of Compliance Rate (CR) and MSE for TITSP, Time-LLM, Qwen4MTS, UniTime, and Llama-3.1-8B across various instructed actions with highlighted best (in **bold**) and second-best (in _underlined_) results.
>
> | **Instruction**                 | **TITSP (CR)** | **TITSP (MSE)** | **Time-LLM (CR)** | **Time-LLM (MSE)** | **Qwen4MTS (CR)** | **Qwen4MTS (MSE)** | **UniTime (Qwen) (CR)** | **UniTime (Qwen) (MSE)** | **Llama-3.1-8B (CR)** | **Llama-3.1-8B (MSE)** |
> |---------------------------------|----------------|-----------------|-------------------|--------------------|-------------------|--------------------|-------------------------|-------------------------|------------------------|------------------------|
> | Linear Growth and Linear Decay  | **0.83**       | **1.15**        | 0.38             | 3.45               | _0.69_            | _1.90_             | 0.54                   | 2.73                   | 0.32                  | 4.95                  |
> | Linear Growth and Linear Decay  | **0.79**       | **1.17**        | 0.49             | 2.85               | **0.79**          | _1.34_             | 0.57                   | 2.28                   | 0.41                  | 2.80                  |
> | Linear Trend Up                 | _0.90_         | **1.03**        | 0.63             | 1.71               | 0.76              | _1.08_             | 0.63                   | 1.65                   | **0.91**              | 1.15                  |
> | Linear Trend Down               | **0.87**       | **0.88**        | 0.64             | 1.55               | 0.71              | 1.36               | 0.51                   | 1.59                   | _0.85_                | _0.92_                |
> | Exponential Growth              | **0.89**       | **1.33**        | 0.58             | 2.59               | _0.63_            | _2.07_             | 0.60                   | 2.38                   | 0.58                  | 2.35                  |
> | Exponential Decay               | **0.84**       | **1.25**        | 0.56             | 2.26               | 0.67              | 2.10               | _0.69_                 | _2.05_                 | 0.46                  | 2.39                  |
> | Keep Stable                     | **0.98**       | _0.35_          | 0.76             | 0.76               | 0.93              | 0.48               | 0.83                   | 0.62                   | _0.95_                | **0.33**              |
> | Decrease Amplitude              | **0.90**       | _0.91_          | 0.85             | 1.04               | **0.90**          | **0.84**           | 0.79                   | 1.09                   | 0.52                  | 1.89                  |
> | Increase Amplitude              | **0.94**       | **0.94**        | 0.79             | 1.20               | _0.89_            | _0.96_             | 0.81                   | 1.03                   | 0.75                  | 1.35                  |
> | Logarithmic Growth              | _0.77_         | _1.65_          | 0.49             | 2.31               | **0.79**          | **1.55**           | 0.60                   | 1.73                   | 0.55                  | 1.94                  |
> | Logarithmic Decay               | **0.83**       | **1.68**        | 0.48             | 2.19               | _0.81_            | _1.69_             | 0.67                   | 2.04                   | 0.63                  | 2.60                  |

---

> ### Author Response · Authors · 2024-11-21
> **Response to Reviewer GGqR**
>
> Dear reviewer: I wonder if our response can solve your concerns! Thank you!

---

> > ### Comment · Reviewer_GGqR · 2024-11-26
> >
> > I appreciate the authors' responses, which partially address my concerns. However, I believe the writing quality of this paper does not meet the standards expected for this conference. I encourage the authors to review some of the recent papers they have cited, such as:
> >
> > - Jin, M., Wang, S., Ma, L., Chu, Z., Zhang, J. Y., Shi, X., ... & Wen, Q. (2023). Time-llm: Time series forecasting by reprogramming large language models. arXiv preprint arXiv:2310.01728.
> > - Liu, Y., Hu, T., Zhang, H., Wu, H., Wang, S., Ma, L., & Long, M. (2023). itransformer: Inverted transformers are effective for time series forecasting. arXiv preprint arXiv:2310.06625.
> >
> > These papers exemplify the level of clarity and structure that is expected. I recommend the authors consider these examples to improve the organization and presentation of their work. Consequently, I have decided to maintain my original score.

---

### Official Review · Reviewer_mT1k · 2024-11-03

**Soundness:** 3
**Presentation:** 2
**Contribution:** 3
**Rating:** 5
**Confidence:** 3

**Summary:**

The article describe a new model to incorporate textual information with a more traditional timeseries forecasting model. It does so by combining an embedding computed from the historical numerical data with an embedding computing from the textual information. The combined embedding is then used to generate the forecast.

The model is tested both on real-world data, where it shows competitive results, and on generated data, where it is shown to follow the instructions included in the textual information.

**Strengths:**

1. It is good that zero shot examples of descriptions which have not been provided in the training set have been tested with. Without those, the narrow set of possible descriptions could have made it impossible to check whether the result quality came from the model overfitting on these descriptions or not.
2. Training the model using generated data and computing how well the model follows the instructions is a relatively clean way to do a proof of concept of the idea, which is appropriate currently, as the field of using LLM and timeseries models together is still in its infancy.

**Weaknesses:**

1. There seems to be a mismatch between the described technique used to apply the modification (equation 3), and the examples shown (figure 3). According to the equation, the data in the forecast window should be a pure affine function, without any of the noise shown in figure 3.
2. While the model is tested against other multimodal text+timeseries models, it should also be tested against pure LLM approaches: just plugging the text and the history in a prompt for GPT 4 or LLama 3, and looking at the generated output. While such an approach won't scale to long series, recent work have shown it to be surprisingly decent at forecasting under textual instructions. See: LLM Processes by Requiema 2024 for a slightly more complex approach, but there may be more appropriate references for the more direct one.
3. Hyperparameters and training curiculum for the timeseries portion of the model are missing.

**Questions:**

1. For table 4, can you provide the same results, but for your model instead of only for TimeLLM? It would make it more obvious whether your model succeed on those tasks with incorrect textual information.
2. For real world dataset, was the textual information always constant (as shown in section B.3) for each dataset? This would allow a finetuned model to fully ignore it, since it could bake said information in its weights anyway.

---

> ### Author Response · Authors · 2024-11-18
> **Response To Reviewer mT1k**
>
> ### Thank you for your precious comments!
> The following are our responses to your concerns.
>
> ---
>
> ### Comment 1:
> *There seems to be a mismatch between the described technique used to apply the modification (equation 3), and the examples shown (figure 3). According to the equation, the data in the forecast window should be a pure affine function, without any of the noise shown in figure 3.*
>
> **Response:**
> We thank the reviewer for highlighting this point. Equation (3) indeed describes a pure affine function; however, to ensure an increasing trend in certain time series, we allowed the slope \(A\) to vary within the forecast window. This deliberate choice introduces some noise, as shown in Figure 3, but it demonstrates the model’s ability to adapt to evolving trends. For clearer examples without slope variation, please refer to Figure 10 in the Appendix (page 20). We have clarified this in the revised manuscript on page 4, where a comment is added to clarify this important point.
>
> ---
>
> ### Comment 2:
> *While the model is tested against other multimodal text+timeseries models, it should also be tested against pure LLM approaches: just plugging the text and the history in a prompt for GPT-4 or Llama 3, and looking at the generated output. While such an approach won't scale to long series, recent work has shown it to be surprisingly decent at forecasting under textual instructions. See: LLM Processes by Requiema 2024 for a slightly more complex approach, but there may be more appropriate references for the more direct one.*
>
> **Response:**
> We thank the reviewer for this suggestion. Although LLMs have shown some capability with time series data, they are fundamentally designed for language tasks and often struggle with numerical accuracy, as highlighted by several studies. This limitation motivated our dual-channel approach, where time series and text are processed in specialized frameworks, leveraging an expert model for each modality.
>
> In *Table 2 (page 9)*, we conduct an experiment by directly prompting Llama-3.1-8B-Instruct to perform these tasks. The results show good understanding of simple instructions but significant failures in most tasks. This approach also demonstrates instability, as the output may be challenging to directly utilize due to the mixture of numerical values and textual content. The designed prompt is shown in *Appendix I*.
>
> In the Appendix (see *Table 5*), we present an experiment comparing our dual-channel method with GPT4TS—a purely LLM-based model for time series (for descriptive text instead of instructions). Despite GPT’s strong backbone (compared to Qwen used for our approach), our method outperforms it, confirming that dual-channel designs are more effective for multimodal tasks. Additionally, as the reviewer suggested, we conducted a new experiment focused on instruction-based tasks, which is the main focus of the paper. Here, our model also demonstrated superior compliance rates compared to Qwen4TS, underscoring the advantages of dual-channel methods for instruction-based text. These results are now included in the revised manuscript in *Table 2 (page 9)*.
>
>
> ---
>
> ### Comment 3:
> *Hyperparameters and training curriculum for the timeseries portion of the model are missing.*
>
> **Response:**
> We thank the reviewer for pointing this out. The missing experimental details regarding the hyperparameters and training curriculum for the time series feature extractor are now included in the updated version of the manuscript. These details are provided in *Section I of Appendix*, where we outline the specific settings and the training procedure used for this part of the model, as well as the experimental setup for training UniTime and Qwen4MTS for the new additional experiments.

---

> ### Author Response · Authors · 2024-11-18
> **Response To Reviewer mT1k - question parts**
>
> ### Question 1:
> *For Table 4, can you provide the same results, but for your model instead of only for TimeLLM? It would make it more obvious whether your model succeeds on those tasks with incorrect textual information.*
>
> **Response:**
> We thank the reviewer for this insightful suggestion. As part of our ongoing experiments, we aim to address this by evaluating our model under the same conditions. Specifically, for the same base time series (same context length), we provide multiple different instructions and observe that our model achieves a high compliance rate, demonstrating its ability to follow instructions accurately. In contrast, TimeLLM exhibits lower compliance, highlighting the importance of the instructions. We appreciate the reviewer’s input, and we have now included these results in the updated manuscript for even more models (Qwen4TS and UniTime) (see *Table 2, page 9*).
>
> ---
>
> ### Question 2:
> *For the real-world dataset, was the textual information always constant (as shown in Section B.3) for each dataset? This would allow a fine-tuned model to fully ignore it, since it could bake said information in its weights anyway.*
>
> **Response:**
> We thank the reviewer for raising this important point. In our experiments, the format of the textual prompts varied across datasets, ensuring that the model was exposed to different types of instructions and did not simply memorize a single format. However, within each dataset, the prompt format remained consistent to ensure a fair evaluation of the model's ability to handle the specific instructions. This approach prevents the model from "baking" the textual information into its weights and ensures it adapts to diverse instructions. We have clarified this in *Section B.3, page 19* of the updated manuscript, where we add another prompt format for an additional dataset (Traffic).

---

> ### Author Response · Authors · 2024-11-18
> **Response To Reviewer mT1k - result table**
>
> For convenience, we also provide table 5 in the paper about test results
>
> ### Table: Comparison of Compliance Rate (CR) and MSE for TITSP, Time-LLM, Qwen4MTS, UniTime, and Llama-3.1-8B across various instructed actions with highlighted best (in **bold**) and second-best (in _underlined_) results.
>
> | **Instruction**                 | **TITSP (CR)** | **TITSP (MSE)** | **Time-LLM (CR)** | **Time-LLM (MSE)** | **Qwen4MTS (CR)** | **Qwen4MTS (MSE)** | **UniTime (Qwen) (CR)** | **UniTime (Qwen) (MSE)** | **Llama-3.1-8B (CR)** | **Llama-3.1-8B (MSE)** |
> |---------------------------------|----------------|-----------------|-------------------|--------------------|-------------------|--------------------|-------------------------|-------------------------|------------------------|------------------------|
> | Linear Growth and Linear Decay  | **0.83**       | **1.15**        | 0.38             | 3.45               | _0.69_            | _1.90_             | 0.54                   | 2.73                   | 0.32                  | 4.95                  |
> | Linear Growth and Linear Decay  | **0.79**       | **1.17**        | 0.49             | 2.85               | **0.79**          | _1.34_             | 0.57                   | 2.28                   | 0.41                  | 2.80                  |
> | Linear Trend Up                 | _0.90_         | **1.03**        | 0.63             | 1.71               | 0.76              | _1.08_             | 0.63                   | 1.65                   | **0.91**              | 1.15                  |
> | Linear Trend Down               | **0.87**       | **0.88**        | 0.64             | 1.55               | 0.71              | 1.36               | 0.51                   | 1.59                   | _0.85_                | _0.92_                |
> | Exponential Growth              | **0.89**       | **1.33**        | 0.58             | 2.59               | _0.63_            | _2.07_             | 0.60                   | 2.38                   | 0.58                  | 2.35                  |
> | Exponential Decay               | **0.84**       | **1.25**        | 0.56             | 2.26               | 0.67              | 2.10               | _0.69_                 | _2.05_                 | 0.46                  | 2.39                  |
> | Keep Stable                     | **0.98**       | _0.35_          | 0.76             | 0.76               | 0.93              | 0.48               | 0.83                   | 0.62                   | _0.95_                | **0.33**              |
> | Decrease Amplitude              | **0.90**       | _0.91_          | 0.85             | 1.04               | **0.90**          | **0.84**           | 0.79                   | 1.09                   | 0.52                  | 1.89                  |
> | Increase Amplitude              | **0.94**       | **0.94**        | 0.79             | 1.20               | _0.89_            | _0.96_             | 0.81                   | 1.03                   | 0.75                  | 1.35                  |
> | Logarithmic Growth              | _0.77_         | _1.65_          | 0.49             | 2.31               | **0.79**          | **1.55**           | 0.60                   | 1.73                   | 0.55                  | 1.94                  |
> | Logarithmic Decay               | **0.83**       | **1.68**        | 0.48             | 2.19               | _0.81_            | _1.69_             | 0.67                   | 2.04                   | 0.63                  | 2.60                  |

---

> ### Author Response · Authors · 2024-11-21
> **Response To Reviewer mT1k**
>
> Dear reviewer: I wonder if our response can solve your concerns! Thank you!

---

> > ### Comment · Reviewer_mT1k · 2024-11-21
> >
> > Thanks for answering my comments and questions.
> >
> > W1: Figure 10 seems to indicate that "Linear Growth" is gotten by adding an affine function to the original data. This may be compatible with equations (4) and (5) (which are not very clear), but is definitely still not compatible with equation (3) and Figure 3 (which is not compatible with such a transformation).  Please make sure that the method you used to modify your data is accurately documented in your paper to allow other researchers to reproduce your work.
> >
> > W2: Thanks for adding the extra experiments. Larger scale LLMs would have performed better, but would have been more costly.
> >
> > W3: Thanks for adding the extra details.
> >
> > Q1: While Table 8 does show the impact of changing the way the textual information is phrased (and shows that it has an impact on the model), it doesn't outright give incorrect information (as in Table 4). I would still be curious to see the result of such an experiment for your model.
> >
> > Q2: Is the model trained with all the datasets at once, or one version is trained for each dataset? (This may already been mentioned in the paper.) It is true that varied prompts for each dataset would help in the former case, it wouldn't have an impact in the later case.
> >
> > Overall, I would still need to think whether to increase your paper score or keep it as is. I will have to take some time to reread my fellow reviewers comments and reread the paper before doing so.

---

### Author Response · Authors · 2024-11-18

# Dear Reviewers

We would like to express our sincere gratitude for your time and valuable feedback on our paper. Below, we provide a detailed response to each of the reviewers' comments and outline the revisions made to address their concerns. We have followed your suggestions and believe the manuscript has significantly improved as a result.

## Summary of Changes

In response to the reviewers' comments, we have made the following key changes to the manuscript:

- Clarification of the dataset generation to handle the question on the apparent mismatch between Equation 3 and Figure 3 on **page 4** (**Reviewer mT1k**).
- Additional experiments by adding two baselines (UniTime and Qwen pure LLM) for text instruction in Table 2 on **page 9** (**Reviewers mT1k, GGqR, YdJR, and We4d**).
- Incorporation of additional state-of-the-art methods, including other multimodal papers such as UniTime, on **page 3** (**Reviewers We4d and GGqR**).
- Detailed architecture design (number of layers, architecture, hyperparameters) for the time series portion, as well as a detailed description of the datasets in **Section I of the appendix** (on **page 30**) (**Reviewers mT1k and GGqR**).
- Clarification about AutoPrompter on **page 5** (**Reviewer GGqR**).
- **Added more related work** on traditional time-series prediction models in **Section 2.1**.

The detailed responses to each reviewer’s comments are provided below.

---

### Meta-Review · Area_Chair_rCos · 2024-12-20

**Metareview:**

This paper presents a text-instruction following time-series forecasting model that uses LLMs to combine embeddings from the historical numerical data with embeddings from the textual information. While the problem space of integrating text and time-series is very timely and practical, several reviewers expressed concerns (that the AC agrees with) the rather simplistic textual instructions and evaluation methodology, and the limited real-world practical applicability of modeling textual inputs in terms of such clear, simple instructions. While the authors do also experiment with descriptive text prompts in the appendix, this seems more of an afterthought and seems disconnected from the main theme of the paper. I would urge the authors to resubmit the paper to a future venue, after performing a more comprehensive evaluation using both descriptive text prompts and a richer, noisier set of textual instructions.

**Additional Comments On Reviewer Discussion:**

Some reviewers asked for adding more baselines and multimodal approaches to the evaluation. Furthermore, reviewers had several questions on the dataset generation, and the potential of leakage in the evaluation methodology and metrics. The authors addressed the authors these points satisfactorily. However they were not able to alleviate the concerns of some reviewers around the limited applicability of the instruction-following setting and the lack of a richer set of text prompts in the evaluation framework.

---

### Decision · Program_Chairs · 2025-01-22

Reject